

# A Level 3 Monthly Gridded Ice Cloud Dataset Derived from a Decade of CALIOP Measurements

David Winker[1], Xia Cai[2, 1], Mark Vaughan[1], Anne Garnier[2, 1], Brian Magill[2, 1], Melody Avery[1], and Brian Getzewich[1]

[1]NASA Langley Research Center, Hampton, Virginia 23681, USA
[2]Analytical Mechanics Associates, Hampton, Virginia 23666, USA

**Correspondence:** David Winker (david.winker@nasa.gov)

**Abstract.** Clouds play important roles in weather, climate, and the global water cycle. The Cloud-Aerosol Lidar with Orthogonal Polarization (CALIOP) onboard the Cloud-Aerosol Lidar and Infrared Pathfinder Satellite Observation (CALIPSO) spacecraft has measured global vertical profiles of clouds and aerosols in the Earth's atmosphere since June 2006. CALIOP provides vertically resolved information on cloud occurrence, thermodynamic phase, and properties. We describe Version 1.0
of a monthly gridded ice cloud product derived from over ten years of global, near-continuous CALIOP measurements. The primary contents are monthly, vertically resolved histograms of ice cloud extinction coefficient and ice water content (IWC) retrievals. The CALIOP Level 3 Ice Cloud Product Version 1.0 is built from the CALIOP Version 4.20 Level 2 5-km Cloud Profile product, which, relative to previous versions, features substantial improvements due to more accurate lidar backscatter calibration, better extinction coefficient retrievals, and a temperature-sensitive parameterization of IWC. The gridded ice cloud
data are reported as histograms, which provides data users with the flexibility to compare CALIOP's retrieved ice cloud properties with those from other instruments with different measurement sensitivities or retrieval capabilities. It is also convenient to aggregate monthly histograms for seasonal, annual or decadal trend and climate analyses. This CALIOP gridded ice cloud product provides a unique characterization of the global and regional vertical distributions of optically thin ice clouds and deep convection cloud tops, and should provide significant value for cloud research and model evaluation. A DOI has been issued
for the product: https://doi.org/10.5067/CALIOP/CALIPSO/L3_ICE_CLOUD-STANDARD-V1-00 (Winker et al., 2018).

## 1 Introduction

Covering a large fraction of the globe, atmospheric ice clouds have significant impacts on Earth's radiation budget and also play a key role in the atmospheric hydrologic cycle. Due to the cold temperatures at which ice clouds are found, they impact
both longwave and shortwave radiation, with the net balance dependent on optical depth and other cloud properties (Berry and Mace, 2014; Hong et al., 2016). Deep convective clouds contain large amounts of ice but represent a very small fraction of global cloud cover (Sassen et al., 2009). Ice detrained from deep convection and in situ formation within moist layers in the





upper troposphere are responsible for most of the global coverage of ice cloud. Most of this global coverage is optically thin, making a small contribution to the global ice mass budget but significant contributions to the radiation budget (Haladay and
Stephens, 2009).

Satellite sensors are our only means of observing the global distribution of ice clouds and characterizing their properties. These global observations are essential for understanding the mean distribution of atmospheric ice and its variability. Accurate representation of ice clouds is important for both numerical weather prediction and climate modeling. Satellite observations are critical for assessing whether these models produce realistic simulations of atmospheric ice clouds, both for simulating
a realistic energy balance and for properly modeling the hydrologic cycle. Indeed, Waliser et al. (2009) pointed out that discrepancies between models are much larger for ice water path (IWP) and ice water content (IWC) than for parameters such as global mean cloud cover, for which there are better observational constraints. Intercomparison studies, however, persistently show a large spread in IWP between satellite datasets (Waliser et al., 2009; Eliasson et al., 2011; Duncan and Eriksson 2018). Satellite datasets exhibit similar geographical patterns of ice distribution but there are large differences in the magnitude of IWP.
These differences between observational datasets make it difficult to validate models and to identify avenues for improvement.

Global data on IWP have been available for decades from a number of passive visible and infrared (VIS/IR) satellite sensors and several passive microwave sensors (Bühl et al., 2017) but any one sensor is sensitive to only part of the IWP column (Eliasson et al., 2011; Waliser et al., 2009). VIS/IR sensors are only sensitive to thin clouds and the upper portions of deep clouds while nadir-viewing microwave sensors can retrieve ice through thick clouds but have trouble detecting thin ice clouds. These
varying sensitivities are one reason for the large differences between satellite ice cloud datasets. Passive nadir-viewing sensors cannot measure the profile of IWC but only column IWP. Limb-viewing instruments such as the Microwave Limb Sounder (MLS; Waters et al., 2006; Wu et al., 2008) and the SubMilllimeter Radiometer (SMR) on the Odin satellite (Murtagh et al., 2002) provide vertically-resolved profiles of ice in the upper troposphere but have poor horizontal resolution, and interpretation of the measurements is complicated by the long tangent path through the atmosphere (Wu et al., 2009).

While passive imagers and radiometers provide detailed cloud mapping from space, a deeper and more comprehensive understanding of the spatial and temporal distributions of clouds on a global scale requires knowledge of cloud vertical distributions and multi-layer occurrence frequencies. New capabilities for retrieving vertically resolved IWC became available with the launch of the Cloud-Aerosol Lidar and Infrared Pathfinder (CALIPSO; Winker et al., 2010) and CloudSat (Stephens et al., 2002) satellites in 2006. The Cloud-Aerosol Lidar with Orthogonal Polarization (CALIOP), the CALIPSO lidar (Hunt et al.,
2009; Winker et al., 2009) operates at 532 nm and 1064 nm and has high sensitivity to optically thin ice clouds which are often undetected by the CloudSat W-band (94 GHz) profiling radar due to their small particle sizes (Mace et al., 2009). However, the CloudSat radar can penetrate optically thick clouds and all but the densest convective systems and therefore adds observations of dense ice clouds where CALIOP signals are completely attenuated.

Currently, perhaps the most complete observations of IWC throughout the vertical column come from two datasets which
combine collocated data from CALIPSO and CloudSat: 2C-ICE (Deng et al., 2010) and DARDAR (Delanoë and Hogan, 2010). Noel et al. (2018) studied the representivity of the sun-synchronous observations from CALIOP relative to observations over the diurnal cycle from the CATS lidar. In some cases CALIOP observations represented extreme values of the diurnal cycle of

cloud profiles, while taking CALIOP observations from both local overpass times (0130 and 1330) provided a good indication
of the daily average cloud fraction profile, over both ocean and land. We have constructed a lidar-only Level 3 Ice Cloud
Product based on the CALIPSO Version 4.2 Level 2 Cloud Profile product which is more continuous, covering both day and
night from June 2006 through December 2016. When data processing artifacts caused by intermittent low-energy laser pulses
during the later years of the mission are resolved (Tackett et al., 2022) the product will be extended to cover the full CALIPSO
mission. In addition to the more complete temporal coverage than the radar-lidar products, the dataset also benefits from using
the latest versions of the CALIPSO cloud extinction and cloud thermodynamic phase algorithms (Young et al., 2018; Avery et
al., 2020).

The primary contents of the CALIPSO Level 3 Ice Cloud Product (hereafter, L3-ICE) are monthly statistics on ice cloud
extinction and IWC. Results are reported on a uniform three-dimensional global grid of $2.5°$ longitude by $2.0°$ latitude and 120
m vertical resolution, from the sea level surface to 20.2 km altitude. For each month, three data files are created. One reports
statistics exclusively for daytime measurements; a second reports statistics exclusively for night-time measurements; and the
third reports the combined day and night statistics.

Previous studies show that comparison of the mean values of the various satellite IWC datasets are difficult to interpret
because instruments have different sensitivities and observe different portions of the IWP column (Waliser et al., 2009; Li
et al., 2016). Intercomparison of histograms can be more meaningful and can identify differences in instrument sensitivities
(Duncan and Eriksson 2018). Therefore, L3-ICE profiles of ice cloud extinction and IWC are presented as gridded monthly
histograms. The histograms are constructed using sample counts, rather than normalized frequency values, to allow proper
aggregation of statistics to larger spatial and/or temporal scales.

The remainder of this paper provides a detailed introduction to the product, including the method of construction, quality
control measures, characteristics of the data, and uncertainties. Section 2 discusses the CALIOP Level 2 5-km Cloud Profile
product on which the L3-ICE is based. Section 3 describes the methods used to select high confidence ice cloud extinction and
IWC data and aggregate this Level 2 data onto a three-dimensional global grid. Section 4 presents a few results to illustrate
product contents. Section 5 discusses sources of uncertainty (uncertainties due to sparse sampling, inability to probe deep
convection, Level 2 cloud clearing). Section 6 assesses L3-ICE strengths and weaknesses via comparisons with the DARDAR
and 2C-ICE products. Finally, Section 7 presents a summary and a few thoughts on future development.

## 2 Input data

CALIOP is an elastic backscatter lidar transmitting linearly polarized laser pulses at 532 nm and 1064 nm. CALIOP is nadir
viewing with a 90-meter diameter receiver footprint every 335 meters creating a curtain of profile observations along the
satellite track. Backscattered light from the CALIOP laser is detected and sampled at high vertical resolution. The 532 nm
receiver separately measures backscattered light polarized parallel and perpendicular to the polarization of the outgoing beam,
allowing the identification of cloud thermodynamic phase (Hu 2007). Below an altitude of 8.2 km, profiles are sampled at a
vertical resolution of 30 m and every profile is downlinked. Between altitudes of 8.2 km and 20.2 km, profiles are averaged





onboard the satellite to 60 m vertical and 1 km horizontal resolution before being downlinked (Hunt et al., 2009). Details of radiometric calibration and other Level 1 processing are described in Powell et al. (2009), Kar et al. (2018), Getzewich et al. (2018), and Vaughan et al. (2019). Strongly scattering cloud and aerosol layers can be detected from single return profiles but averaging of multiple lidar shots is required to detect optically thin layers (Winker et al., 2009). Therefore, CALIOP Level 2

processing employs an iterative multi-scale averaging scheme to detect both weakly and strongly scattering layers at the highest practical horizontal resolution (Vaughan et al., 2009). This multi-scale averaging scheme results in a collection of atmospheric features detected at horizontal resolutions ranging from 1/3 km to 80 km. Detected features are then classified as aerosol or cloud (Liu et al., 2019) and cloud layers are classified as liquid or ice. Ice layers are further classified as randomly oriented ice (ROI) or horizontally oriented ice (HOI) using differences in the backscatter and depolarization signatures of the layers (Avery

et al., 2020).

L3-ICE is built from the Version 4.20 Level 2 Cloud Profile Product (hereafter L2-CPro). L3-ICE uses L2-CPro altitude-resolved profiles of cloud properties, including 532 nm extinction coefficients and IWC, from the sea-level surface to 20.2 km altitude, which are reported at a vertical resolution of 60 m. Due to signal-to-noise ratio (SNR) limitations in the highest resolution data, profiles of particulate extinction are only retrieved for clouds detected at horizontal averaging resolutions of 5

km, 20 km, and 80 km (Young and Vaughan 2009; Young et al., 2018).

Profiles of IWC ($\mathrm{g\,m^{-3}}$) are derived from ice cloud extinction coefficients using a temperature-dependent parameterization based on in situ measurements acquired during a number of aircraft field campaigns conducted between 1991 and 2007 (Heymsfield et al., 2014; hereafter H14):

$$IWC(z) = (\frac{\rho}{3})\sigma(z)\alpha_T e^{\beta_T T(z)} \tag{1}$$

where $\sigma(z)$ and $T(z)$ are the ice cloud extinction coefficient ($\mathrm{m^{-1}}$) and temperature (°C), respectively, at altitude $z$. The value adopted for the density of ice, $\rho$, is $0.91\,\mathrm{g\,cm^{-3}}$ (Heymsfield et al., 2014). The temperature-dependent fitting parameters $\alpha_T$ and $\beta_T$, given in Table 1, were determined from least squares fitting of volume extinction coefficients and IWC measured in situ. Of several fits to the data explored in H14, Equation 1 is the piece-wise fit that best reproduces the aircraft data across the full temperature range of $-86$°C to $0$°C. The temperature dependence of the IWC vs. extinction relationship can be interpreted

as due to a broadening of the particle size distribution as temperature increases (H14).

**Table 1.** Temperature-dependent fitting parameters $\alpha_T$ and $\beta_T$ used in Equation 1.

| Temperature (T), °C | $\alpha_T$ | $\beta_T$ |
|---|---|---|
| -56 < T < 0°C | 308.4 | 0.0152 |
| -71 < T < -56°C | $9.1774 \times 10^4$ | 0.117 |
| -85 < T < -71°C | 83.3 | 0.0184 |

L3-ICE also uses ancillary data on atmospheric state and surface elevation which is contained in L2-CPro. Data describing the atmospheric state includes temperature, pressure, relative humidity, and tropopause height, all taken from the NASA Global



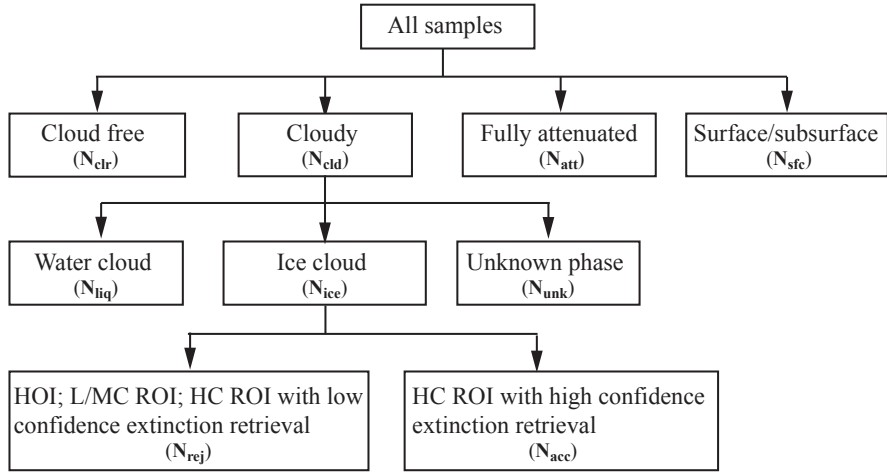

**Figure 1.** Decision tree for scene classification leading to selection of high confidence ROI samples having high-confidence extinction retrievals, ($N_{acc}$).

Modeling and Assimilation Office MERRA-2 product (Gelaro et al., 2017). Surface elevation data is taken from a Digital Elevation Model (DEM) developed by the CloudSat team (Tanelli et al., 2014) which is primarily based on data from the NASA Shuttle Radar Topography Mission (SRTM) (NASA JPL 2013), augmented by surface elevation data from GTOPO30, ASTER-GDEM, and NSIDC at high latitudes. Complete contents of the CALIOP layer and profile products are given in Vaughan et al. (2023).

## 3 Method

The L3-ICE is derived from the L2-CPro product in a series of steps involving data selection, quality screening, and construction of the histograms. Flags contained in L2-CPro are used to identify the locations of ice clouds within the profile. Quality screening is then applied to identify ice cloud layers which have high confidence extinction retrievals. Finally, quality-screened monthly statistics are aggregated onto a global three-dimensional (3-D) grid, in the form of histograms of ice cloud extinction and IWC, along with various types of sample counts. Each of these steps is described in detail in the sections that follow.

### 3.1 Selection of Ice Cloud Layers

Figure 1 outlines the decision tree used to select ice cloud data for inclusion in L3-ICE and steps in the process are illustrated in Figure 2. During this process several types of sample counts are accumulated and reported in L3-ICE (Table 2). The standard 532 nm attenuated backscatter browse image in Figure 2 (a), in which Level 1 profiles are averaged to 5 km horizontal resolution, shows a scene dominated by optically thin ice clouds above an altitude of about 8 km. Clouds below 8 km are predominantly supercooled water and are mostly opaque to CALIOP. The *Atmospheric_Volume_Description* (AVD) parameter



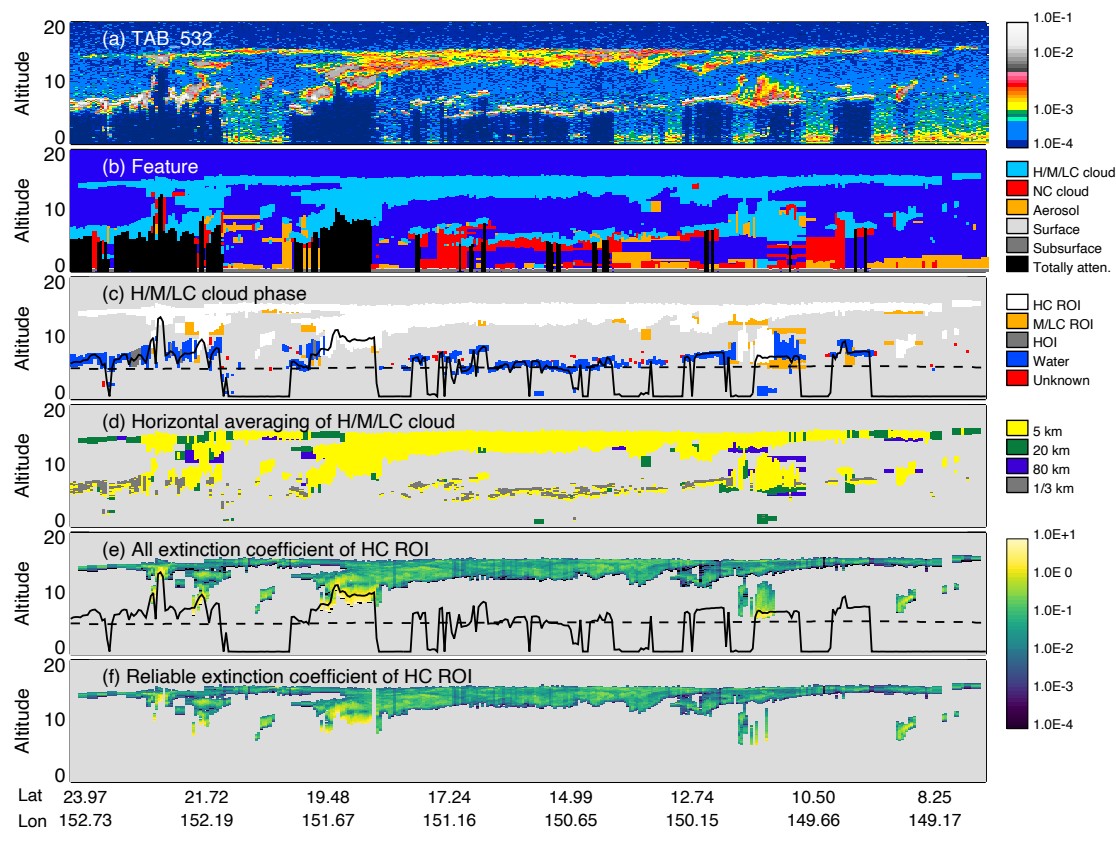

**Figure 2.** Illustration of the steps involved in the selection of high confidence ice clouds and extinction values: a) 532 nm attenuated backscatter profiles ($\mathrm{km}^{-1}\mathrm{sr}^{-1}$); b) feature classifications; c) cloud phase classifications; d) horizontal averaging required to detect cloud layers; e) profiles of all extinction coefficients ($\mathrm{km}^{-1}$) retrieved within high confidence ROI cloud layers; f) extinction coefficient profiles after quality screening. Solid and dashed black lines in panels (c) and (e) show the altitude where the overlying optical depth reaches 2 and the freezing level, respectively. The scene is taken from granule 2008-07-28-T15-38-54ZN.

contains feature classification flags for each range bin in the L2-CPro profiles. The AVD contains flags identifying feature type and, for cloudy bins, the cloud type, cloud phase, and the level of confidence in the discrimination of cloud from aerosol and in the identification of cloud thermodynamic phase (Vaughan et al., 2023). As shown in Figure 1, the first step of the Level 3 ice cloud selection process uses AVD flags to identify range bins which are clear, cloudy, totally attenuated, contain the surface return or are below the surface. Sample counts of each feature type are accumulated in the parameters $N_{clr}$, $N_{cld}$, $N_{att}$, and

$N_{sfc}$, respectively. Range bins designated as "clear" are cloud-free but may contain aerosol.

Figure 2 (b) illustrates the scene shown in Figure 2 (a) after feature classification. Black areas indicate the lidar signal has been completely extinguished by optically thick overlying clouds. The AVD also contains a quality assurance (QA) flag which indicates the confidence in the cloud-aerosol classification performed by the cloud-aerosol discrimination (CAD) algorithm (Liu et al., 2009). Cloudy range bins classified with high, middle, or low confidence are shown in light blue. Range bins

classified as "No Confidence Cloud" are shown in red. A classification of "No Confidence" often indicates erroneous detection rather than a true cloud and these bins are rejected from further consideration.

Next, Ice-Water Phase flags in the AVD are used to identify cloudy bins as ice cloud, liquid cloud, or "unknown phase" in cases where the ice-water phase algorithm was unable to classify the cloud phase. This happens most often when scattering from optically thin clouds is very weak or when very low in-layer SNR leads to contradictory results within the phase algorithm.

These sample counts are accumulated, respectively, in the parameters $N_{ice}$, $N_{liq}$, and $N_{unk}$, where $N_{cld} = N_{ice} + N_{liq} + N_{unk}$. The AVD cloud phase flag further identifies whether the clouds are composed of randomly oriented ice (ROI) particles or horizontally oriented ice (HOI) particles. Figure 2 (c) shows the classification of cloud phase in the example scene, separately showing the occurrence of ROI identified with high confidence (HC ROI, in white), with medium or low confidence (orange), HOI (gray), and liquid (blue). The phase flags show the upper layer is cirrus composed of ROI, classified with high confidence.

Clouds located between 5 and 9 km altitude in this scene are mostly liquid clouds. Figure 2 (d) shows the horizontal resolutions at which the cloud layers in Figure 2 (c) are detected. Most of the ice clouds are reported at 5 km horizontal resolution, with averaging over 20 km required to detect the optically thinner parts of the cirrus layer, while many of the water clouds are detected with single shots. The solid black line shows the altitude at which the overlying cloud optical depth (the cloud optical depth integrated from 20.2 km down to altitude $z$) reaches 2. Because substantial aerosol layers are only rarely lofted to cirrus

altitudes, we ignore aerosol contributions in these optical depth calculations. When the column cloud optical depth is less than 2 the line drops to the surface. In this scene the optical depth 2 threshold is most often exceeded when a liquid cloud is encountered. The horizontal dashed line indicates the altitude of the freezing level, showing that most of the liquid clouds in this scene are supercooled. Note that in this scene there are no high confidence ROI layers detected beneath liquid clouds.

**Table 2.** Vertically resolved sample counts which are reported in the product.

| Sample counts | Symbol |
|---|---|
| Surface or subsurface samples | $N_{sfc}$ |
| Totally attenuated samples | $N_{atten}$ |
| Cloud-free clear air samples | $N_{clr}$ |
| Total cloud samples | $N_{cld}$ |
| Liquid cloud samples | $N_{liq}$ |
| Unknown phase cloud samples | $N_{unk}$ |
| Total ice cloud samples | $N_{ice}$ |
| Ice cloud samples, rejected | $N_{rej}$ |
| Ice cloud samples, accepted | $N_{acc}$ |



## 3.2 Tests to ensure high quality extinction retrievals

After range bins containing ice clouds are identified, several tests are applied to ensure that only range bins containing high quality ice cloud extinction retrievals are selected for inclusion in the product. These screening steps are described below. Sample counts of ice cloud bins which pass all these screening tests are counted in *Ice_Cloud_Accepted_Samples* ($N_{acc}$), and samples which fail any of these tests are counted in *Ice_Cloud_Rejected_Samples* ($N_{rej}$), so that $N_{ice} = N_{acc} + N_{rej}$.

High confidence ROI test. Accurate extinction retrievals require a good estimate of the particle extinction-to-backscatter ra-
tio (the "lidar ratio"). It is not uncommon for plate-like crystals to have a quasi-horizontal orientation. Specular lidar backscatter from these oriented crystals is much higher than from the more common randomly oriented crystals, and these oriented crystals can be identified by their anomalously high backscatter and near-zero depolarization (Sassen 1977; Nöel and Sassen 2005). Lidar ratios of layers containing HOI crystals are highly variable because the volume lidar backscatter is very sensitive to the relative concentrations of ROI and HOI. The result is that extinction retrievals of clouds containing HOI are highly uncertain
(Mioche et al., 2010). Retrievals of layers without HOI but having low or medium phase confidence are also uncertain. Therefore, the AVD *Cloud_Phase* and *Phase_QA* flags are used to select only those samples identified as high confidence ROI. The rejected HOI and low or medium confidence ROI sample counts are accumulated in $N_{rej}$.

Extinction quality control (QC) test. The outcome of each extinction retrieval is indicated in Level 2 products by the *Extinction_QC_532* flag. Only retrievals with *Extinction_QC_532* values of 0, 1, 2, 16, and 18 (see Table 3) are accepted for
L3-ICE. Results from retrievals having other *Extinction_QC_532* values are rare but are excluded because they indicate either a failed retrieval or a retrieval which is likely to include erroneous values (Young et al., 2018). Table 3) lists the frequency with which these extinction QC values occur in Level 2 cloud data. In July 2008, roughly 80% of extinction retrievals performed on semi-transparent ice clouds with the remaining retrievals performed on opaque clouds. A complete listing of all extinction QC flags assigned by the Version 4.20 algorithm is given in Table 2 of Young et al.(2018).

**Table 3.** Description of extinction quality control flags and their frequency of occurrence for ice cloud retrievals in July 2008 in the Version 4.20 L2-CPro.

| QC flag | Interpretation | Frequency |
|---------|---------------|-----------|
| 0 | Unconstrained retrieval; initial lidar ratio unchanged | 0.616 |
| 1 | Constrained retrieval; solution constrained by measured two-way transmittance | 0.191 |
| 2 | Unconstrained retrieval; initial lidar ratio reduced to achieve successful solutions for backscatter coefficients and uncertainties | 0.007 |
| 16 | Feature identified as opaque, initial lidar ratio unchanged | 0.074 |
| 18 | Feature identified as opaque; initial lidar ratio reduced to achieve successful solutions for backscatter coefficients and uncertainties | 0.112 |
| Others | Unsuccessful retrievals of cloud extinction | <0.001 |

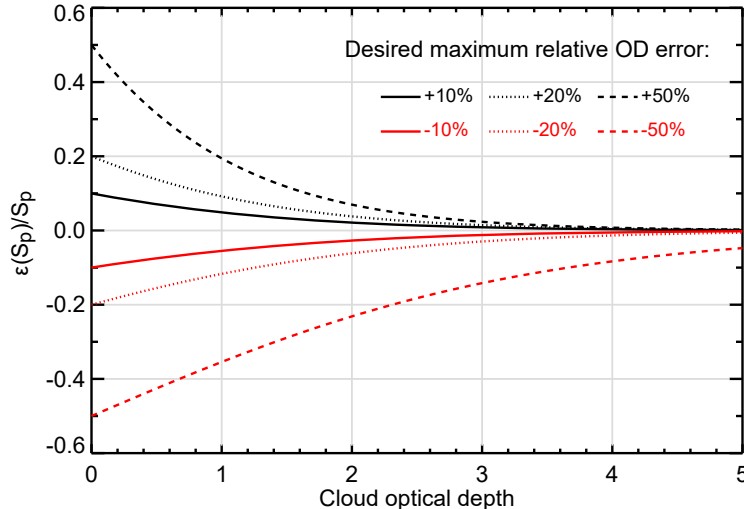

**Figure 3.** Lidar ratio accuracy required to achieve optical depth accuracy of 10%, 20% or 50%, based on Eq. 41 in Young et al., (2013).

Extinction coefficient uncertainty. The uncertainty of each retrieved extinction value is estimated by the extinction retrieval algorithm and reported in L2-CPro. Divergence of the extinction uncertainty profile indicates a failed retrieval and range bins at and below the point of divergence are assigned an extinction uncertainty of 99.9 $km^{-1}$. These range bins are excluded, as are all bins at lower altitudes in the profile, since extinction solutions at lower altitudes depend critically on the extinctions retrieved in layer above.

Accepted extinction range. Extinction values outside the range $-0.1\,km^{-1}$ to $10.0\,km^{-1}$ are considered suspicious. In weakly scattering layers, signal noise can produce negative attenuated backscatter values, resulting in negative extinction coefficients and IWC values. Ignoring these negative values will result in a positive bias when computing means and medians. Therefore, a lower extinction limit of $-0.1\,km^{-1}$ is used to retain negative extinction values resulting from signal noise while excluding large negative outliers resulting from erroneous retrievals. When cloud extinction is as large as $10.0\,km^{-1}$ the signal

is attenuated very rapidly and retrieval uncertainties become large. From Equation 1, the maximum IWC corresponding to an extinction value of $10.0\,km^{-1}$ is roughly $1.0\,g\,m^{-3}$.

    Ice clouds beneath water clouds. While it is common for precipitating ice to occur below supercooled water cloud layers (Zhang et al., 2010; Silber et al., 2021), fewer than 2% of the ice layers detected by CALIOP are found beneath supercooled water layers. This is in part because most supercooled water layers are opaque to CALIOP. Retrievals of extinction profiles and

optical depths of water clouds are only performed on layers averaged horizontally to 5-km or more. These retrievals are highly uncertain due to the difficulties of accounting for the effects of multiple scattering (Young et al., 2013) and also from averaging over cloud which is horizontally inhomogeneous. Because these retrieval uncertainties propagate downward into the retrievals of underlying layers, retrievals of ice clouds beneath liquid clouds are highly uncertain. Therefore, the relatively few ice layer retrievals from beneath supercooled water layers are ignored.



Overlying optical depth threshold filter. As cloud optical depth increases, extinction retrievals become increasingly sensitive to errors in the lidar ratio used in the retrieval (Young et al., 2013). Under the condition of perfect calibration, constant scattering ratio within the cloud, and negligible multiple scattering, an exact expression for the relative error in retrieved cloud optical depth due to uncertainty in the particulate lidar ratio, $S_p$, is given by Equation 41 in Young et al. (2013). Based on this equation, Figure 3 shows the relative accuracy with which $S_p$ must be estimated to retrieve cloud optical depth with a relative error of

10%, 20%, or 50%. As seen in the figure, as optical depth approaches zero the relative optical depth error approaches the relative lidar ratio error: $\varepsilon(\tau_p)/\tau_p = -\varepsilon(S_p)/S_p$. Note that the required accuracy depends on the sign of the error and becomes increasingly asymmetric at larger optical depths.

    Retrieving optical depths larger than 2 with even 50% accuracy requires unreasonable accuracy in the lidar ratio used. Further, attenuation of the lidar backscatter signal significantly reduces the SNR. After penetrating an optical depth of 2 the

attenuated backscatter signal is attenuated to less than 2% of the unattenuated magnitude. Therefore, samples with overlying cloud optical depth larger than 2 are rejected and only ice cloud samples from the top two optical depth of the column are included in L3-ICE. Garnier et al. (2021) present statistics showing that, during daytime, nearly all ice clouds penetrated by the lidar signal have optical depths less than 2 and, at night, optical depths less than 3. Thus this filter mostly removes uncertain retrievals within opaque ice cloud layers.

Panels (e) and (f) of Figure 2 show ice cloud extinction coefficient profiles before and after screening for high quality extinction retrievals. Lines indicating overlying optical depth of 2 and the freezing level are the same as in panel (c). Comparison of panels (e) and (f) shows the most significant impact of applying quality filters is the exclusion of bins deep within opaque cloud layers where the overlying optical depth exceeds 2, such as near latitudes 19.0° N and 11.0° N.

## 3.3   Product Contents

**Table 4.** Histogram bin ranges for ice cloud extinction coefficients and IWC. The bin number is calculated with Equations 2 and 3.

| Bin number | $\sigma$ bin boundaries, $\mathrm{km}^{-1}$ | IWC bin boundaries, $\mathrm{g\,m}^{-3}$ |
|---|---|---|
| 1 | $(-3.401 \times 10^{+38}, -1.0 \times 10^{-1})$ | $(-3.401 \times 10^{+38}, -1.0 \times 10^{-2})$ |
| 2-16 | $[-1.0 \times 10^{-1}, -1.0 \times 10^{-4})$ | $[-1.0 \times 10^{-2}, -1.0 \times 10^{-5})$ |
| 17 | $[-1.0 \times 10^{-4}, 0.0)$ | $[-1.0 \times 10^{-5}, 0.0)$ |
| 18 | $[0.0, +1.0 \times 10^{-4})$ | $[0.0, +1.0 \times 10^{-5})$ |
| 19-43 | $[+1.0 \times 10^{-4}, +1.0 \times 10^{+1})$ | $[+1.0 \times 10^{-5}, +1.0 \times 10^{0})$ |
| 44 | $[+1.0 \times 10^{+1}, +3.402 \times 10^{+38})$ | $[+1.0 \times 10^{0}, +3.402 \times 10^{+38})$ |

CALIOP L3-ICE files are generated in Hierarchical Data Format 4 (HDF4) by the CALIPSO data management team and publicly distributed by NASA's Atmospheric Science Data Center (ASDC; https://asdc.larc.nasa.gov/project/CALIPSO/ CAL_LID_L3_Ice_Cloud-Standard_V1-00) and by the AERIS/ICARE data center (https://www.icare.univ-lille.fr/data-access/ data-archive-access/?dir=CALIOP/CAL_LID_L3_Ice_Cloud.v1.00/). Complete listings of all scientific data sets (SDSs) and



metadata reported in these files are given in Appendix A and B. The primary contents of L3-ICE are histograms of ice cloud
532 nm extinction coefficients and IWC, and the associated gridded monthly sample counts (Table 2). Data are reported on a
uniform three-dimensional global grid of 2.5° longitude by 2.0° latitude and 120 m vertical resolution. Extinction coefficients
and IWC values of samples passing all the quality tests described above are aggregated into vertically resolved histograms
and the sample counts listed in Table 2 are reported for each grid cell. The structure of the histograms is described in Table
4. Bins 2-43 contain extinction coefficient values between $-0.1\,\mathrm{km}^{-1}$ and $10.0\,\mathrm{km}^{-1}$. The corresponding IWC values range
from $-0.01\,\mathrm{g\,m}^{-3}$ to $1.0\,\mathrm{g\,m}^{-3}$. Retrieved values outside this range are flagged as outliers and, as a diagnostic, are reported in
bins 1 and 44. Bins 17 and 18 contain samples near zero, with absolute magnitude less than $10^{-4}\,\mathrm{km}^{-1}$ and $10^{-5}\,\mathrm{g\,m}^{-3}$ for
extinction and IWC respectively.

To span the large range in extinction coefficient ($\sigma$) and IWC, the histograms are defined using logarithmic bin boundaries.
In log space, the size of bins 2-43 is 0.2, thus five bins represent 1 order of magnitude. The definition of the bin boundaries is
given by:

$$\text{Bin number of } \sigma = \begin{cases} 1 & -3.401 \times 10^{+38} < \sigma < -1.0 \times 10^{-1}, \\ floor\left(\dfrac{(-1.0) - log_{10}|\sigma|}{0.2} + 2\right) & -1.0 \times 10^{-1} \leq \sigma < -1.0 \times 10^{-4}, \\ 17 & -1.0 \times 10^{-4} \leq \sigma < 0, \\ 18 & 0 \leq \sigma < +1.0 \times 10^{-4}, \\ floor\left(\dfrac{log_{10}\sigma - (-4)}{0.2} + 19\right) & +1.0 \times 10^{-4} \leq \sigma < +1.0 \times 10^{+1}, \\ 44 & +1.0 \times 10^{+1} \leq \sigma < +3.402 \times 10^{+38} \end{cases} \tag{2}$$

$$\text{Bin number of } IWC = \begin{cases} 1 & -3.401 \times 10^{+38} < IWC < -1.0 \times 10^{-2}, \\ floor\left(\dfrac{(-2.0) - log_{10}|IWC|}{0.2} + 2\right) & -1.0 \times 10^{-2} \leq IWC < -1.0 \times 10^{-5}, \\ 17 & -1.0 \times 10^{-5} \leq IWC < 0, \\ 18 & 0 \leq IWC < +1.0 \times 10^{-5}, \\ floor\left(\dfrac{log_{10}IWC - (-5)}{0.2} + 19\right) & +1.0 \times 10^{-5} \leq IWC < +1.0 \times 10^{0}, \\ 44 & +1.0 \times 10^{0} \leq IWC < +3.402 \times 10^{+38}. \end{cases} \tag{3}$$

Weighted summing over these histograms gives the monthly mean extinction (in $\mathrm{km}^{-1}$) and IWC (in $\mathrm{g\,m}^{-3}$) in each 3D
grid cell. The total number of valid ice cloud samples within a 3D grid cell is $N_{acc}$:

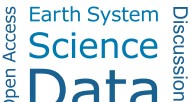

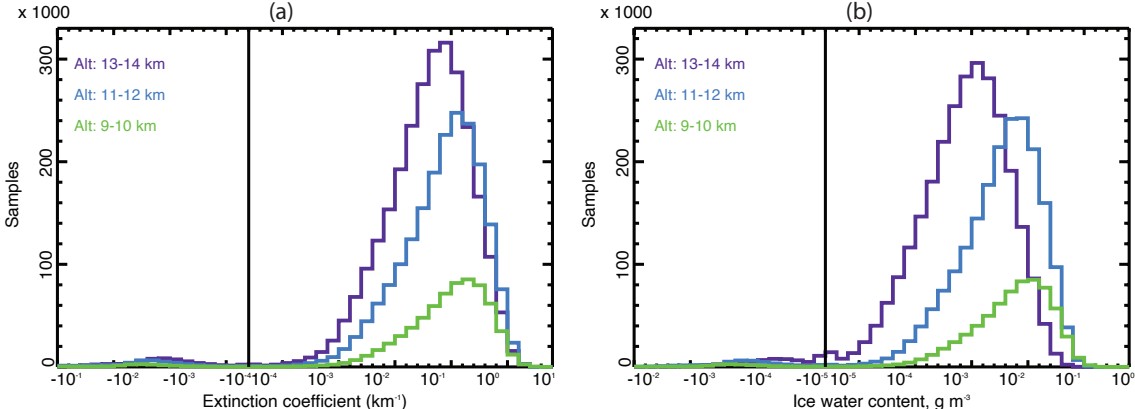

**Figure 4.** Histograms of night-time ice cloud 532 nm channel extinction coefficients (a) and IWC (b) observed at three altitudes in the tropics (23.5° S - 23.5° N) in July 2008. The $X$-axis is a split logarithmic scale to show both positive and negative values. The vertical solid line separates negative and positive values.

$$N_{acc} = \sum_{i=1}^{44} N_{acc,i} \qquad (4)$$

where $N_{acc,i}$ is the number of accepted ice cloud samples in bin $i$ of a histogram.

Figure 4 shows extinction coefficient and $IWC$ histograms for three altitude ranges in the tropics in July 2008. The mode of the distributions moves to larger values as the altitude decreases and there are many more ice cloud samples detected in the highest altitude region between 13 and 14 km than in the lowest altitude region between 9 and 10 km. As histograms, even in log-space, tend to be skewed, medians are provided as a second measure of central tendency of the values. The differences of mean and median values (Table 5) are an indication of skewness and for some applications median values are more meaningful than averages. The median ice cloud extinction coefficient and IWC in each 3D grid cell are reported in the *Extinction_Coefficient_532_Median* and *Ice_Water_Content_Median* parameters.

**Table 5.** Mean and median values of histograms in Figure 4.

|  | Extinction coefficient, $km^{-1}$ | | | IWC, $g\,m^{-3}$ | | |
|---|---|---|---|---|---|---|
| Altitude | 13-14 km | 11-12 km | 9-10 km | 13-14 km | 11-12 km | 9-10 km |
| Mean | 0.2891 | 0.4391 | 0.5386 | 0.006145 | 0.02041 | 0.03108 |
| Median | 0.1292 | 0.2048 | 0.3246 | 0.002048 | 0.008155 | 0.01292 |

The number of profiles acquired in each grid cell, over land and over ocean, are reported in the parameters *Land_Surface_Samples* and *Water_Surface_Samples*. The sum of these two parameters represents the total number of 5-km profiles acquired within each monthly grid cell. Ancillary data on atmospheric state and surface elevation are also included in the product. Surface ele-





**(a) L2 Aerosol/Cloud Profile**

Vertical resolution: 60 m
Vertical range: -0.5 km ~ 30.1 km

| | |
|---|---|
| +0.16 | ● |
| +0.10 | ● |
| +0.04 | ● |
| -0.02 | ● |
| -0.08 | ● |
| -0.14 | ● |
| -0.20 | ● |
| -0.26 | ● |
| -0.32 | ● |
| -0.38 | ● |
| -0.44 | ● |

**(b) L3 Tropospheric Aerosol**

Vertical resolution: 60 m
Vertical range: -0.5 km ~ 12.0 km

| | |
|---|---|
| +0.16 | ● |
| +0.10 | ● |
| +0.04 | ● |
| -0.02 | ● |
| -0.08 | ● |
| -0.14 | ● |
| -0.20 | ● |
| -0.26 | ● |
| -0.32 | ● |
| -0.38 | ● |

**(c) L3 Ice Cloud**

Vertical resolution: 120 m
Vertical range: -0.5 km ~ 20.2 km



**Figure 5.** Panel (a) shows the vertical grid resolution and vertical range of Level 2 Aerosol/Cloud Profile Products for the lowest altitudes in the profiles. Panels (b) and (c) show the same information, respectively, the Level 3 Tropospheric Aerosol Product and L3-ICE. The numbers on the left column of panel (a) are the lidar altitudes with respect to mean sea level, which are registered to the center of the vertical bins as shown as dots on the right. Note the centers of L3-ICE altitude bins are at the center of every two neighboring vertical bins in the Level 2 Aerosol/Cloud Profile Products and the Level 3 Tropospheric Aerosol Product.

vation data are identical to that in L2-CPro (see Section 2). Gridded atmospheric state data include monthly mean temperature, pressure, and relative humidity, derived from the state data contained in L2-CPro. Full details of product contents can be found in the CALIPSO Data Products Catalog (Vaughan et al., 2023).

The spatial grid of L3-ICE was designed to be compatible with the grid of the Level 3 Tropospheric Aerosol product (Tackett et al., 2018) which is 5.0° longitude by 2.0° latitude and 60 m altitude. The altitude bins for both products are registered to the same lower altitude boundary. In Figure 5, the vertical altitude grid used in L3-ICE is compared to the vertical grids used in L2-CPro and in the Level 3 Tropospheric Aerosol product. In L2-CPro, the AVD is reported at 60 m vertical resolution between 8.2 km and 20.2 km but reported at 30 m vertical resolution below 8.2 km, while the vertical resolution of L3-Ice is

60 m at all altitudes. Using the AVD feature type information, a 60-m range bin below 8.2 km in L3-ICE is defined as cloudy when the feature type of at least one of the two 30-m AVD values is "cloud". If either the upper or lower 30-m bin is classified as ice cloud the entire 60-m grid cell is considered to be an ice cloud, regardless of the classification of the non-ice bin. When aggregating two 60-m bins to one L3 120-m vertical bin, each 60-m cloudy bin is considered as one sample count.

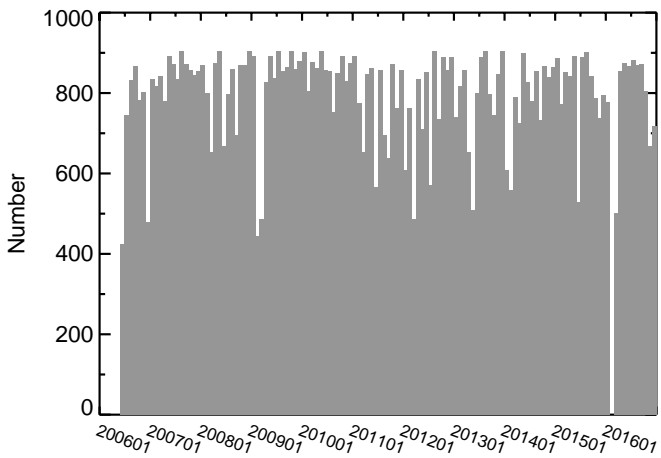

**Figure 6.** The total numbers of processed L2 5 km cloud profile product files, including both daytime and nighttime granules, used in the V1.00 L3-ICE product.

## 4 Data use examples

L3-ICE reports vertically resolved parameters derived from active sensor measurements. Therefore, the design and contents of the product are somewhat, or even substantially, different from commonly used passive sensor data products. This section presents several data usage examples, to illustrate a few characteristics of the product.

### 4.1 Temporal coverage

CALIOP Level 2 data is organized into granules, with each orbit containing one daytime granule and one night granule.
Temporal coverage over the life of the CALIPSO mission is fairly uniform. Figure 6 shows the number of granules used to compute each of the monthly-average Level 3 files from the beginning of data acquisition in June 2006 to the end of 2016. Monthly data coverage varies somewhat month-to-month due to various payload operations which impact data acquisition. For each L3 file, the number of L2 product files used is reported as *Number_of_Level2_Files_Analyzed* in the L3 file metadata. A list of the L2 file names is provided in the *List_of_Input_Files* metadata filed. The largest gaps in sampling are 3 weeks in
February and March 2009, due to the switchover between the primary and backup lasers, and about 45 days from end January until mid-March 2016 caused by a GPS clock problem that affected the entire spacecraft. Detailed information on CALIOP data outages is available on the CALIPSO website at https://www-calipso.larc.nasa.gov/tools/instrument_status/.



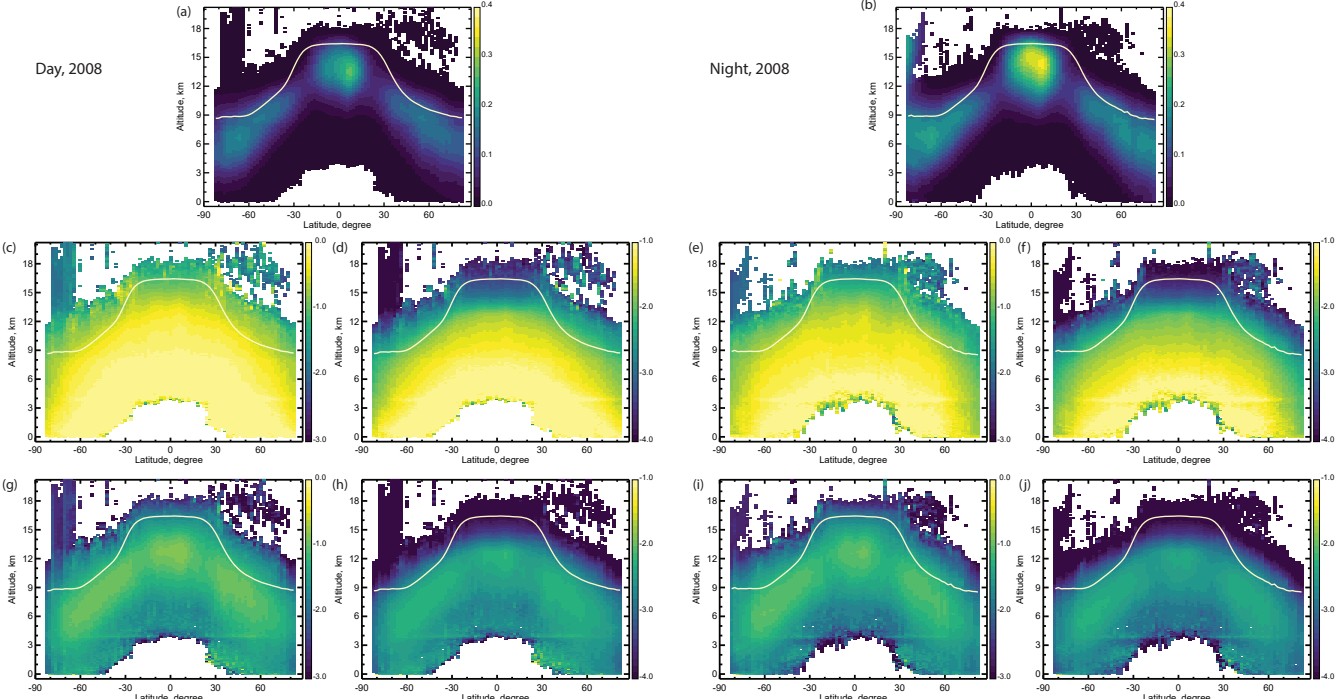

**Figure 7.** Annual zonal mean ice cloud occurrence (a, b), in-cloud extinction coefficient (c, e), in-cloud IWC (d, f), all-sky extinction coefficient (g, i), and all-sky IWC (h, j) for 2008 at the L3-ICE resolution of 2.0° latitude and 120 m altitude. Color bars for extinction coefficeint and IWC are logarithmic. The white line indicates zonal annual mean tropopause height. Left: day; right: night.

## 4.2 Computing height-resolved zonal means

Mean profiles of in-cloud and all-sky IWC can be computed from the L3-ICE parameters *Ice_Water_Content_Histogram*

($N_{acc,i}$) and *Ice_Water_Content_Bin_Boundary* ($IWC_i$) as shown in Equations 5 and 6. In these calculations, the outliers in bins 1 and 44 are excluded from the numerators and the denominator of both equations. In-cloud IWC is computed as:

$$IWC_{in-cloud} = \frac{\sum_{i=2}^{43} N_{acc,i} \times IWC_i}{\sum_{i=2}^{43} N_{acc,i}} \tag{5}$$

where $N_{acc,i}$ and $IWC_i$ are the accepted ice cloud sample count and the mean of the upper and lower bin boundaries for the IWC boundaries of histogram bin $i$ and $N_{acc}$. All-sky IWC can be computed as:

$$IWC_{all-sky} = \frac{\sum_{i=2}^{43} N_{acc,i} \times IWC_i}{N_{cld} + N_{clr}} \tag{6}$$

Figure 7 shows the zonal annual mean occurrence frequency of accepted ice cloud (HC ROI) defined as $N_{acc}/(N_{cld} + N_{clr})$ and the corresponding zonal mean in-cloud IWC and all-sky IWC for day and for night, calculated using Equations 5 and 6.

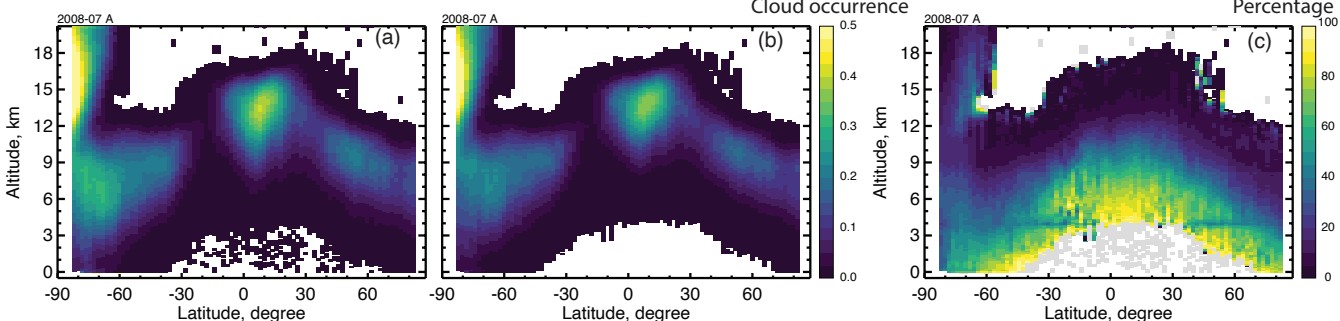

**Figure 8.** Zonal ice cloud occurrence for July 2008 (night) before (a) and after (b) filtering. Cloud occurrence from 0 to 0.5 is coded in color. Panel (c) plots the percentage of samples removed by filtering ($(100 \times N_{rej}/N_{ice})\%$) with gray representing 100% removal.

Zonal patterns of monthly averages are asymmetric about the Equator but annually averaged distributions are quite symmetric. The tropical tropopause layer can be seen as a region of low IWC from 30° S to 30° N and above roughly 14 km. Polar

stratospheric clouds (PSCs) occurring in Antarctic winter can be seen above 12 km at high southern latitudes. PSCs occurring below the 20.2 km upper limit of L3-ICE are included in the product whenever they meet all quality screening requirements.

Figures 7 (d) and (f) show a general pattern of increasing in-cloud IWC as altitude decreases while the all-sky IWC (Figure 7 (h) and (j) shows a rainbow-shaped maximum which varies between 6 km and 12 km with latitude. The decrease in all-sky average IWC below this maximum is due to the increasing frequency of complete attenuation of the lidar signal in optically dense

cloud. Small differences in zonal-mean distributions can be seen between day (left) and night (right). The solar background decreases the daytime CALIOP SNR, degrading detection sensitivity that preferentially affects weakly scattering layers. This leads to cloud occurrence which is somewhat higher at night than during day, and there are corresponding small increases in nighttime IWC relative to daytime.

The same method applies to ice cloud extinction coefficient. To derive the in-cloud and all-sky extinction coefficients,

the *Ice_Water_Content_Histogram* and *Ice_Water_Content_Bin_Boundary* parameters in Equations 5 and 6 are replaced with *Extinction_Coefficient_532_Histogram* and *Extinction_Coefficient_532_Bin_Boundary*.

Figure 8 shows how quality screening applied to the L2-CPro data affects ice cloud occurrence frequencies reported in L3-ICE. Figure 8 (a) shows zonal mean cloud occurrence for July 2008, before quality screening, computed as $N_{ice}/(N_{cld} + N_{clr})$. Figure 8 (b) shows the same data after quality screening, computed as $N_{acc}/(N_{cld} + N_{clr})$. Figure 8 (c) shows the

difference of the unscreened and screened data. Relatively few samples are removed at high altitudes. A large fraction of ice clouds is removed in the lower troposphere due to the increasing uncertainty of retrievals as overlying optical depth increases. The screening completely removes samples at low altitudes in the tropics which were initially classified as low or medium confidence ice clouds at altitudes too warm for ice to occur. A few low confidence samples above the tropical tropopause are also removed.

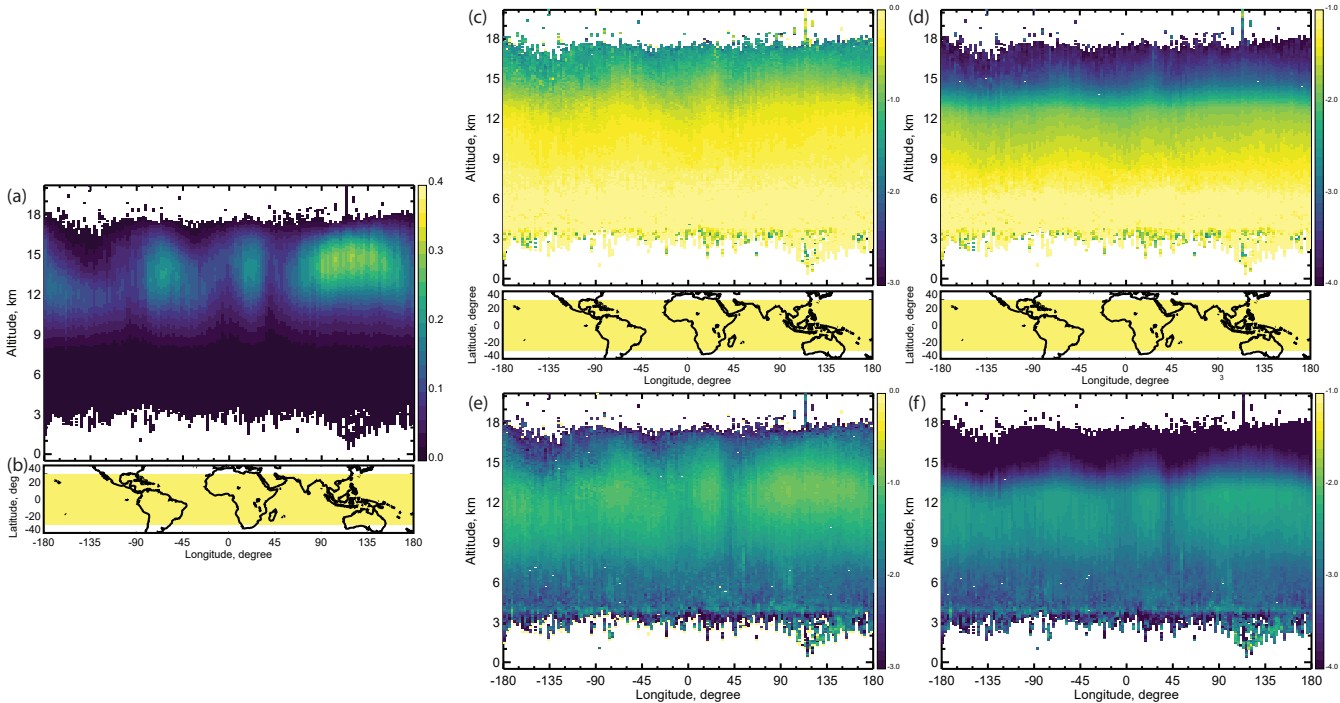

**Figure 9.** Annual-mean meridional distributions between 30° S and 30° N in 2008 (day + night). a) accepted ice cloud occurrence frequency; b) map with yellow shaded area representing tropics between 30° S and 30° N ; c) mean in-cloud extinction coefficient in $\mathrm{km}^{-1}$; d) mean in-cloud IWC in $\mathrm{g\,m}^{-3}$; e) mean all-sky extinction coefficient in $\mathrm{km}^{-1}$; f) mean all-sky IWC in $\mathrm{g\,m}^{-3}$.

### 4.3 Computing meridional means

Figure 9 shows meridional annual average in-cloud and all-sky IWC in 2008 during both day and night between 30° S and 30° N. Means are computed using Equations 5 and 6, as for zonal means. As in the zonal plots, there is a general tendency for in-cloud IWC to increase with decreasing altitude down to the freezing level at about 3 km and for all-sky IWC to have a maximum in the upper troposphere. Frequent occurrence of clouds above 12 km altitude is associated with well-known regions of frequent deep convection in the Western Pacific, the Amazon basin, and central Africa.

### 4.4 Anomaly of ice cloud occurrence and IWC

Figure 10 shows the deseasonalized time series of monthly zonal ice cloud occurrence and all-sky IWC anomaly from 2006 to 2016. The global zonal means (82° N - 82° S) are computed from monthly zonal profile data. A sudden increase in both cloud occurrence and IWC is seen in December 2007 below an altitude of 12 km. This corresponds to a permanent change in the view angle of CALIOP from 0.3° to 3.0° on November 28, 2007 and is related to the detection of HOI particles, which are often found together with ROI within a cloud layer. HOI are readily detected at a view angle of 0.3° but at 3.0° the backscatter returns from HOI are greatly reduced and clouds are more offen classified as ROI. Because cloud layers identified as HOI are

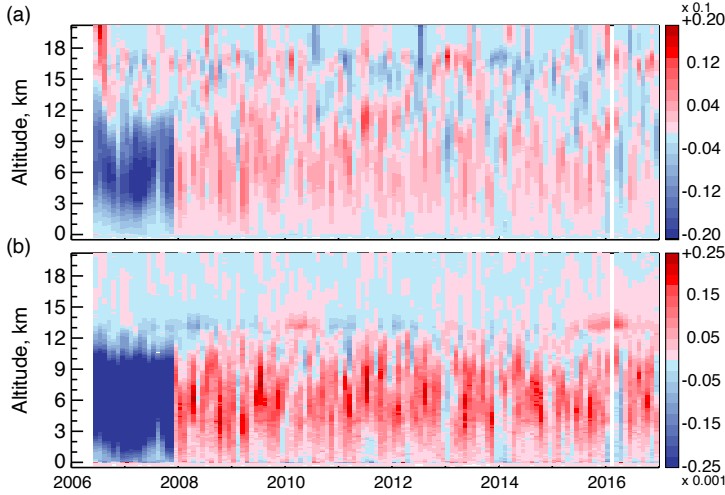

**Figure 10.** Deseasonalized time series of monthly global anomalies of (a) ice cloud occurrence and (b) all-sky IWC from 2006 to 2016. Color coding indicates relative variation from the mean with a range of $\pm 2\%$ for ice cloud occurrence and $\pm 0.25 \times 10^{-3}\,\mathrm{g\,m^{-3}}$ for IWC. The blue regions below 12 km and prior to December 2007 show the impact of the different sensitivities of CALIOP to HOI at view angles of $0.3°$ (prior to December 2007) and at $3.0°$.

removed during quality screening, the occurrence of ROI increases suddenly in December 2007 when fewer clouds containing HOI are identified and removed. Anomalies at the highest altitudes are above the altitude of the tropical tropopause and are driven by the occurrence of polar stratospheric clouds, which tend to occur during polar winter and exhibit large year-year variability (Pitts et al., 2018).

## 5 Uncertainties and biases

As a cloud sensor, CALIOP has the advantages of high detection sensitivity and accurate height determination but the data products are subject to several sources of uncertainty which deserve discussion. In particular, here we focus on incomplete penetration of dense clouds, the nadir-only zero-swath observing geometry, and artifacts due to the way single shot cloud clearing is performed in Version 4.2 Level 2 processing.

### 5.1 Penetration of dense clouds

The CALIOP backscatter signal becomes totally attenuated within optically thick clouds so that only the upper parts of dense clouds are observed. Figure 11 (a) shows the all-sky fraction of Level 2 profiles which reach a given altitude, or the surface, before being completely attenuated, $f_{obs}(z)$:

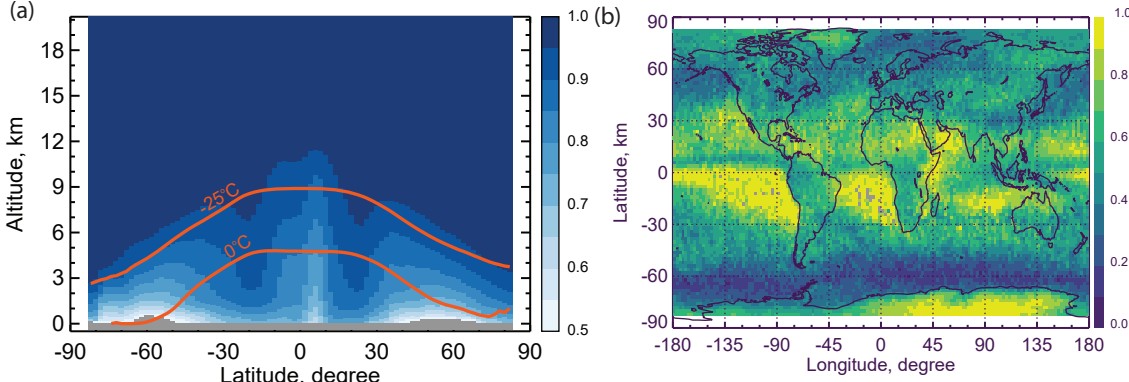

**Figure 11.** (a): Fraction of Level 2 5 km profiles reaching a given altitude before becoming fully attenuated, annual average for 2008; (b) fraction of cloudy profiles where cloud optical depth above the freezing level is less than 2, for 2008.

$$f_{obs}(z) = \frac{1}{N_{tot}} \sum (N_{cld} + N_{clr}) = 1 - \frac{1}{N_{tot}} \sum N_{atten} \tag{7}$$

where $N_{tot} = N_{cld} + N_{clr} + N_{atten}$. It can be seen in Figure 11 (a) that penetration to the freezing level is quite frequent in the tropics and subtropics, except in the core of the Inter-Tropical Convergence Zone (ITCZ). Significant blockage begins to occur at roughly the $-25°C$ level where supercooled water clouds, which tend to be opaque to the lidar signal, begin to occur
frequently.

As discussed earlier, data included in L3-ICE is restricted to samples where the overlying optical depth is less than 2, which sets a limit on the maximum ice water path (IWP) reported in the product. A rough estimate of this limit can be easily derived if all the ice in the column is assumed to be at an altitude corresponding to temperature $T_0$. In that case the maximum value of IWP set by an optical depth limit of $\tau_{max}$ is given by:

$$IWP_{max}(T_0) = \frac{IWC}{\sigma}(T_0) \int \sigma(z)dz = \tau_{max}\frac{IWC}{\sigma}(T_0) \tag{8}$$

Figure 12 plots the temperature-dependent value of $IWC/\sigma$, given by Equation 1, and the corresponding maximum IWP. For $\tau_{max} = 2$, the maximum retrievable IWP is about $200\,\mathrm{g\,m^{-2}}$. The maximum reported IWP decreases with temperature to less than $20\,\mathrm{g\,m^{-2}}$ for very cold clouds found in the tropical tropopause layer, driven primarily by decreasing effective particle size.

For a sense of the extent to which dense clouds impact the ability of CALIOP to detect and retrieve all the ice clouds in the atmospheric column, Figure 11 (b) shows the geographical distribution of the frequency with which the freezing level is reached before penetrating a cloud optical depth of 2. The spatial structure in Figure 11 (b) reflects well-known features of the global cloud distribution, driven by the general circulation of the atmosphere, such as the ITCZ. In subtropical subsidence

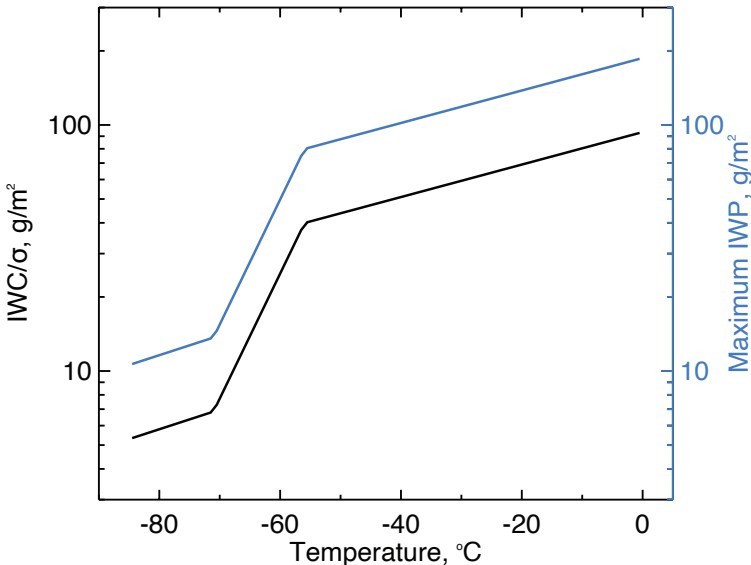

**Figure 12.** Temperature-dependence of the IWC to $\sigma$ ratio used to estimate IWC from retrieved extinction (Equation1, black line) and the limit on IWP due to the requirement that the overlying cloud optical depth is less than 2 (blue line).

regions and over deserts, where there is little convective or mid-level cloud, CALIOP often observes the entire column above
the freezing level. In mid- and high-latitude regions, penetration to the freezing level is blocked by frontal storms and, in fair
weather, can be blocked by supercooled water clouds. This topic will be explored further in Section 6.

**5.2   Sampling considerations**

CALIOP is a nadir-viewing sensor whose measurements are in the form of zero-swath vertical curtains. Relative to passive
imagers, horizontal spatial sampling from CALIOP is very sparse and cloud properties within a grid cell are estimated from
just a few orbit tracks leading to a "representativity uncertainty". Prior to September 2018, when the satellite altitude was
lowered to resume formation flying with CloudSat, the orbit track of CALIPSO was controlled to repeat a fixed pattern of
233 orbits every 16 days. At the Equator, the 233 orbit tracks are spaced by about $1.5°$ longitude and some cells of the often
used $1.0° \times 1.0°$ degree global grid are never sampled. The $2.0° \times 2.5°$ degree lat-lon grid chosen for L3-ICE is a compromise
between high longitudinal resolution and the desire to sample every grid cell. The grid exactly overlaps that of the $2.0° \times 5.0°$
degree lat-lon grid of the Level 3 Aerosol Profile product (Winker et al., 2013) but with twice the longitudinal resolution.

Figure 13 shows the monthly sampling provided by CALIOP. Only 7 or 8 orbit tracks pass through a typical grid cell in one
month, except at high latitudes. Sampling theory can be used to examine uncertainties in sampling areal quantities such as mean
cloud cover from transect measurements (Key, 1993). Results presented in Winker et al. (2017) show that uncertainties due to
the sparse sampling of CALIOP can be reduced significantly by averaging spatially and/or temporally. Kotarba and Solecki
(2021) took a more comprehensive approach using bootstrapped confidence intervals to examine representivity uncertainty

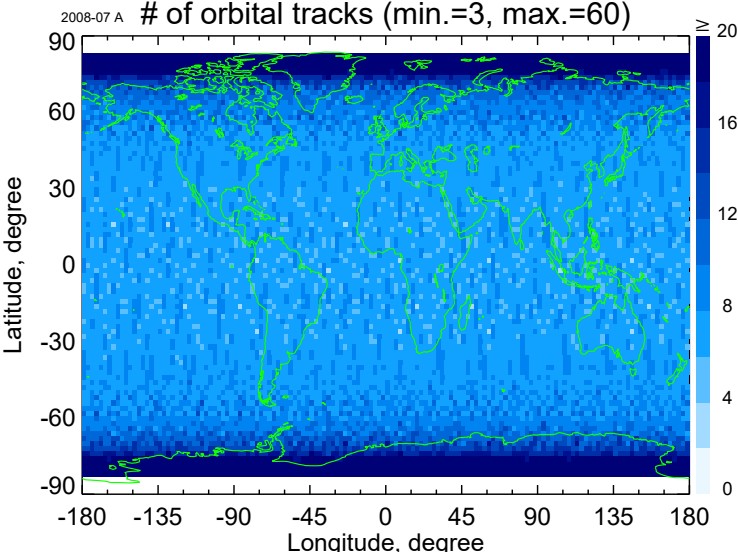

**Figure 13.** Number of tracks per month through a $2.0° \times 2.5°$ degree lat-lon grid in July 2008 (day+night).

in CALIOP estimates of cloud amount. They found that confidence intervals decreased (i.e., improved) in rough proportion to the number of samples when observations were averaged over coarser grids or longer time periods. A subsequent study examining the impacts of representivity uncertainty on the accuracy of mean annual cloud amount, cloud optical depth, and cloud top height found that representivity errors for cloud amount and cloud optical depth behaved similarly with averaging
(Kotarba 2022). For these reasons, data are reported in L3-ICE as sample counts to facilitate proper aggregation to larger and, statistically more meaningful, space-time scales.

To give an observation-based view of the representivity uncertainty due to sparse spatial sampling, 60 consecutive days of L2-CPro data were numbered from 1 to 60. IWP was computed by vertically integrating IWC for each grid cell from even-numbered days and from odd-numbered days. Figure 14 shows distributions of absolute difference $|IWP_{odd} - IWP_{even}|$ and
relative difference $|IWP_{odd} - IWP_{even}|/[0.5(IWP_{odd} + IWP_{even})]$ between IWP from odd and even days at the spatial resolution of L3-ICE (black) and when averaged using a $10° \times 10°$ (red) and $10° \times 20°$ (blue) degree lat-lon grid. The hypothesis is that if a grid cell is well-sampled, the 30-day averaged IWP should be similar whether using even or odd days. The figures show how large differences using the L3-ICE $2.0° \times 2.5°$ degree lat-lon grid can be significantly reduced by averaging to coarser resolutions. Figure 15 shows the greater averaging of zonal means involve enough averaging that odd-day and even-day
$IWP$ agree quite well, using either $2.0°$ or $10°$ latitude increments.

## 5.3 Impact of clearing single-shot ice clouds

The CALIOP Level 2 feature detection algorithm includes a boundary layer cloud-clearing process to mitigate potential cloud contamination when retrieving aerosol optical properties (Vaughan et al., 2005; Vaughan et al., 2009). Data within layers ini-

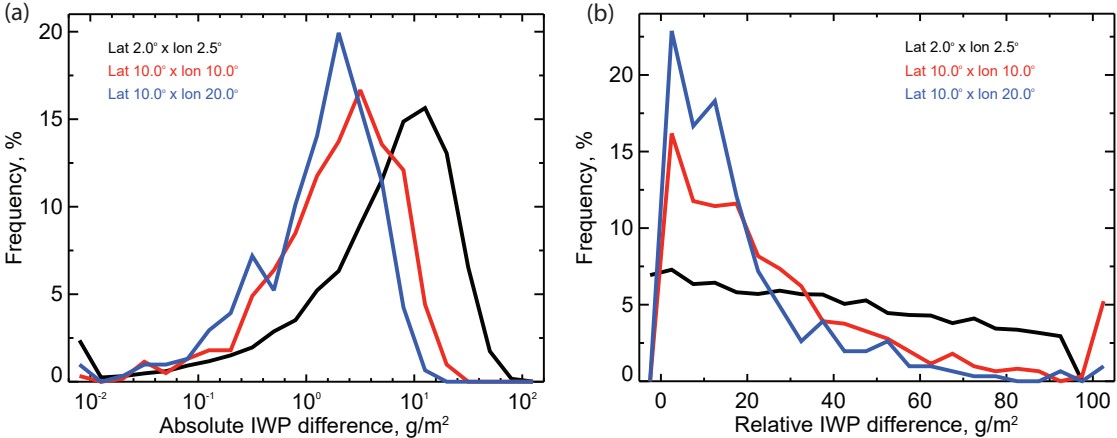

**Figure 14.** Absolute and relative differences between IWP computed from odd and even days using $2.0° \times 2.5°$, $10° \times 10°$, and $10° \times 20°$ degree lat-lon grid cells.

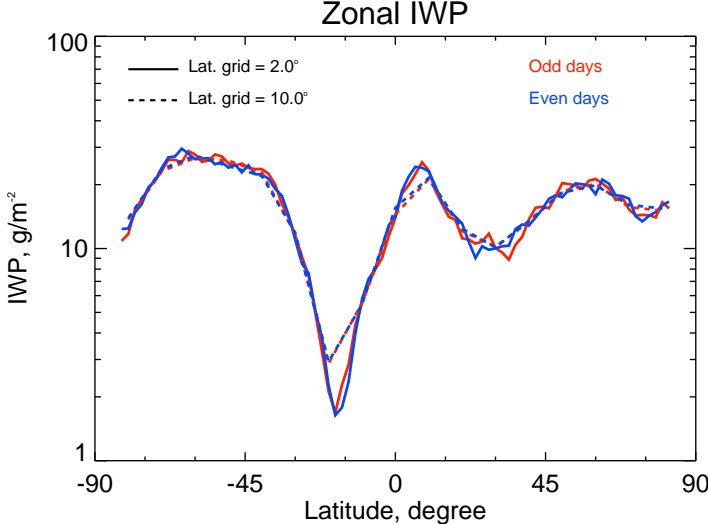

**Figure 15.** Zonal mean IWP computed from 30 odd days (red) and from 30 even days (blue) for latitudinal resolution of $2.0°$ (solid lines) and $10°$ (dashed lines).

tially detected at CALIOP's fundamental 5-km (15 laser shots) along-track resolution are re-examined at single shot resolution.
If clouds with top altitudes at or below 4.0 km are intermittently detected in single shot profiles, the attenuated backscatter data within these clouds are removed and the remaining single shot data are re-averaged to a coarser spatial resolution (i.e., 5-km, 20-km, or 80-km). The homogenized layers detected within these coarser cloud-cleared averages can then be confidently

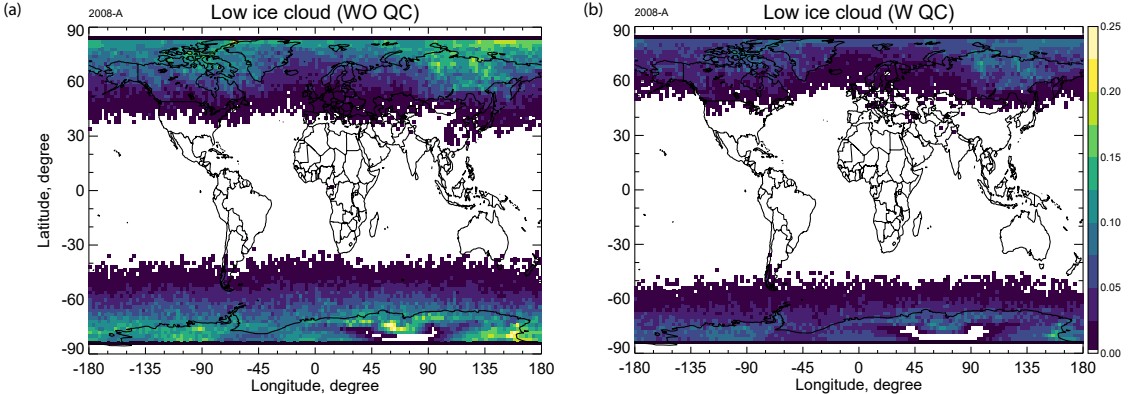

**Figure 16.** a) Annual mean frequency of occurrence of ice clouds below 4.0 km altitude for 2008; b) Fraction of high confidence ROI clouds below 4.0 km with single-shot ice clouds removed.

classified by the CALIOP cloud-aerosol discrimination (CAD) algorithm (Liu et al., 2019). If clouds with tops $\leq$ 4.0 km are detected at single shot resolution in all 15 profiles within a 5-km average, the data removal and re-averaging process is not

executed. Instead, the layer initially detected at 5-km resolution is classified a priori as a cloud, by virtue of having clouds detected within all single shot profiles comprising the 5-km average. Note that clouds detected at single shot resolution with tops above 4.0 km are not subject to the cloud clearing process.

As intended, this clearing process removes strongly scattering water and ice clouds and avoids most of the potential cloud contamination of aerosol in averaged profiles. However, one unforeseen side effect is that the strong scattering from ice clouds

detected at single shot resolution can be removed from the (cloud-cleared) layers subsequently identified as ice clouds in the L2-CPro data. Consequently, biases may exist in the L2-CPro cloud phase assessments, along with the corresponding extinction coefficient and IWC profiles, for low altitude ice clouds detected in polar regions.

The clearing of single-shot features only impacts L2 CPro ice cloud extinction and IWC at high latitudes, as ice clouds are rarely found below 4.0 km altitude at low latitudes. Figure 16 (a) shows the occurrence of ice clouds with cloud tops below 4.0

km is limited primarily to latitudes poleward of 60°. Figure 16 (b) shows fraction of high confidence ROI clouds from which single-shot profiles have been cleared. It can be seen that less than 10% of low altitude 5-km ice clouds are affected by the removal of single-shot layers.

## 6 Comparison with DARDAR and 2C-ICE products

There are currently two radar-lidar products which derive ice cloud extinction and IWC profiles from joint CALIOP and

CloudSat observations using an optical estimation technique: 2C-ICE (Deng et al., 2010; Deng et al., 2015) and DARDAR (Delanoë and Hogan, 2010; Cazenave et al., 2019). The combined capabilities of lidar and 94 GHz cloud profiling radar provide sensitivity to nearly the full spectrum of atmospheric ice, from subvisible cirrus particles to precipitating ice. Additionally, a


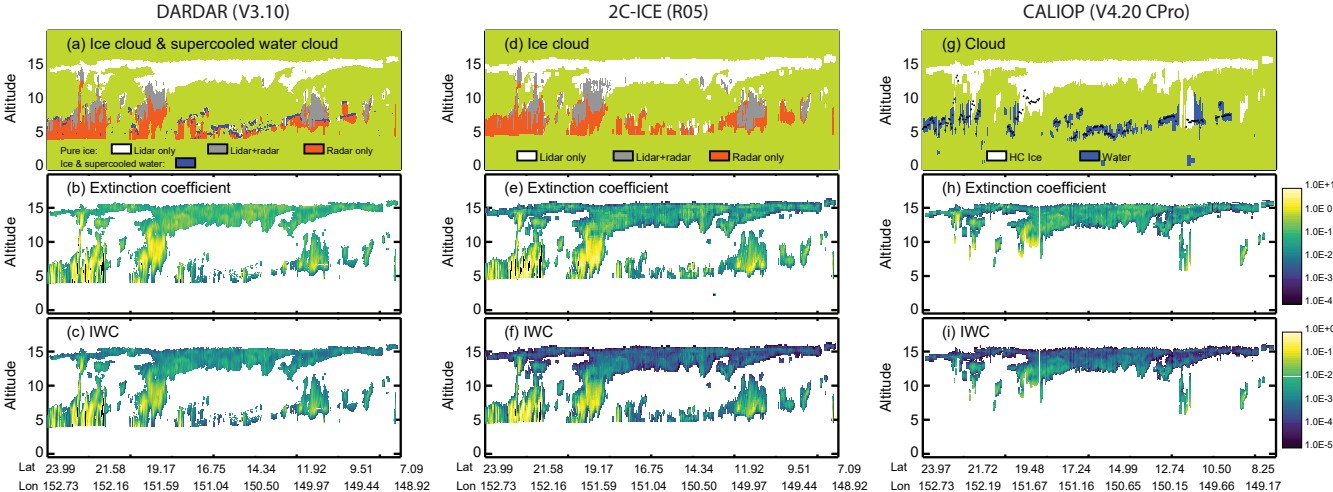

**Figure 17.** Comparison of the DARDAR and 2C-ICE products with L2-CPro, using the filtering of L3-ICE filters, for the same scene as Figure 2. Panels (a), (d) and (g) show results from the three feature masks. Lidar-only, overlap, and radar-only regions are color coded in (a) and (b). Black dots in (g) indicate the level where the overlying optical depth is 2. Ice cloud extinction coefficients ($km^{-1}$) are compared in (b), (e), and (h) and IWC ($g\,m^{-3}$) is compared in (c), (f) and (i). In the extinction and IWC plots, black indicates range bins where the maximum plot value is exceeded.

joint radar-lidar retrieval can, in principle, retrieve vertical profiles of effective particle size and lidar extinction consistent with the observed radar reflectivity and lidar attenuated backscatter. As IWC is a function of effective particle size, this is an

attractive approach to IWC retrievals.

In this section we compare L3-ICE with the 2C-ICE and DARDAR products. We first compare how a selected scene is represented in the three products. Then, to characterize differences at space-time scales of typical interest, we aggregate all available data from 2C-ICE and DARDAR over a month using the same methodology used to construct L3-ICE. These comparisons illustrate similarities and differences a data user might find when using the products at monthly scales. In conducting these

studies, we will restrict our comparisons to those that directly illustrate some salient aspect of the L3-ICE data. Interpretation of differences between DARDAR and 2C-ICE fall outside our scope and have not been pursued.

Figure 17 compares how the scene shown in Figure 2 is reported in the CALIOP products and in the DARDAR and 2C-ICE products. The top row compares the three cloud masks. Figures 17 (h) and (i) are the same as Figures 2 (c) and (f), showing the L2-CPro cloud mask and extinction profiles using the filtering of L3-ICE. The high confidence ice layers (HC ROI) selected

for L3-ICE are shown in white (Figure 17 (g)) and the locations of liquid water clouds, mostly supercooled, are shown in blue. Black dots indicate the altitude where the overlying optical depth equals 2.

Ice clouds composed of small particles are not detected by the radar, while the lower parts of optically thick clouds are not seen by the lidar. A previous study found that the region of radar-lidar overlap is roughly $-50°C$ to $-20°C$ and that less than half of clouds colder than $-50°C$ (above about 12 km in the tropics) are detected by CloudSat, (H14, Figure 1). The



DARDAR and 2C-ICE algorithms are designed to retrieve all three regions – lidar-only, radar-lidar overlap, and radar-only –
       in a consistent manner. In the lidar-only region, a radar reflectivity profile is estimated from the Level 1 lidar profile, based
       on a microphysical model, and a lidar attenuated backscatter profile is estimated in a similar way in the radar-only region. In
       this way, a single retrieval algorithm can be applied in a consistent way throughout the entire depth of the cloud, whether the
       cloud is detected by both radar and lidar or is only detected by one of the instruments. The DARDAR (Figure 17 (a)) and

2C-ICE (Figure 17 (d)) masks are color coded to show the three regions. While the 2C-ICE cloud mask reports only ice clouds,
       between $0°C$ and $-40°C$ the DARDAR cloud mask can report ice, supercooled liquid, or mixed phase if both ice and liquid
       occur within the same measurement volume. The blue areas in Figure 17 (a) indicate either supercooled liquid or mixed-phase
       cloud. As expected, the regions corresponding to radar-only regions are not seen in the L3-ICE mask. Ice shown at lower
       altitudes, down to about 4.0 km, in the DARDAR and 2C-ICE masks is not detected by CALIOP because of blockage by the

supercooled water clouds and, in a few places, by dense overlying ice cloud.

       The second and third rows of Figure 17 compare ice cloud extinction and IWC, respectively. The cirrus layer above 10 km
       is optically thin and fully retrieved in L2-CPro but the overlying optical depth 2 limit is often exceeded within the supercooled
       water layers, blocking CALIOP's view of lower layers. Denser convective cloud is seen near 20° N and 22° N. The two joint
       products retrieve the denser ice in these convective regions, which are not detected by CALIOP due to attenuation within the

overlying cloud. Extinction and IWC retrievals in the upper cloud layer, which is mostly sensed only by lidar, are similar in
       L3-ICE and 2C-ICE but the extinction and IWC retrieved by DARDAR in this region are noticeably higher than the other two
       products.

       To provide a perspective on the ability of lidar to probe dense ice clouds, Figure 18 compares frequency distributions of IWP
       values derived from the three products. Dashed lines show IWP values from L3-ICE and computed from monthly averages of

DARDAR and 2C-ICE IWC profile data using the L3-ICE lat-lon grid. Solid lines show the frequency distributions of IWP
       values computed directly from the Level 2 IWC profile data. Data selection for all cases is the same as that used to produce
       L3-ICE, as described in Section 3.1. The frequency distributions of DARDAR and 2C-ICE are quite similar except for very
       large and very small IWP. As discussed in Section 5.1, the maximum IWP which can be reliably retrieved from CALIOP
       is about 200 $\mathrm{g\,m^{-2}}$. This causes the IWP frequency distribution of L3-ICE to be distorted relative to that of the radar-lidar

products, as CALIOP fails to fully penetrate columns with very high IWP. The frequency distributions in Figure 18 are in
       better agreement for IWP less than 1 $\mathrm{g\,m^{-2}}$, where DARDAR and 2C-ICE rely heavily on the lidar observations. Differences
       between IWP computed from Level 2 IWC data and from gridded monthly-averaged Level 3 IWC data are expected because
       the vertical correlations of cloud overlap are lost in the monthly-averaged data, but these differences are seen to be small for
       all three products.

To give further perspective on differences in the information content of L3-ICE and the two radar-lidar products, Figure 19
       (a)-(c) compare ice cloud occurrence, zonal-mean all-sky profiles of ice cloud extinction, and IWC occurrence in the northern
       tropics from the three products. In these figures we averaged all the available data in each product, except that profile data
       have been screened to remove points which represent fewer than 200 samples. Results for other latitude bands are shown in
       Supplemental Material. The DARDAR and 2C-ICE IWC profiles tend to agree much better with each other than with L3-ICE,

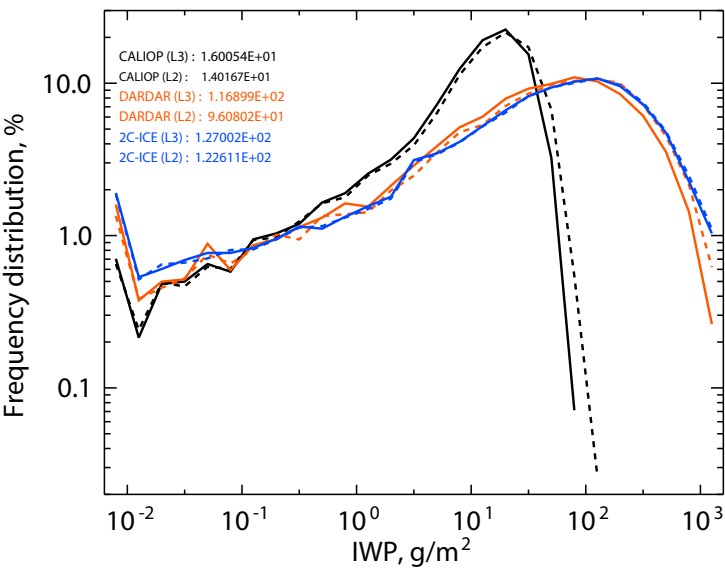

**Figure 18.** Frequency distributions of IWP computed from L3-ICE, DARDAR, and 2C-ICE for all latitudes in July 2008. Solid lines show IWP computed from Level 2 products (along-track curtains). Dashed lines represent IWP computed from monthly grid-cell averages. Legend shows global mean IWP ($\mathrm{g\,m^{-2}}$).

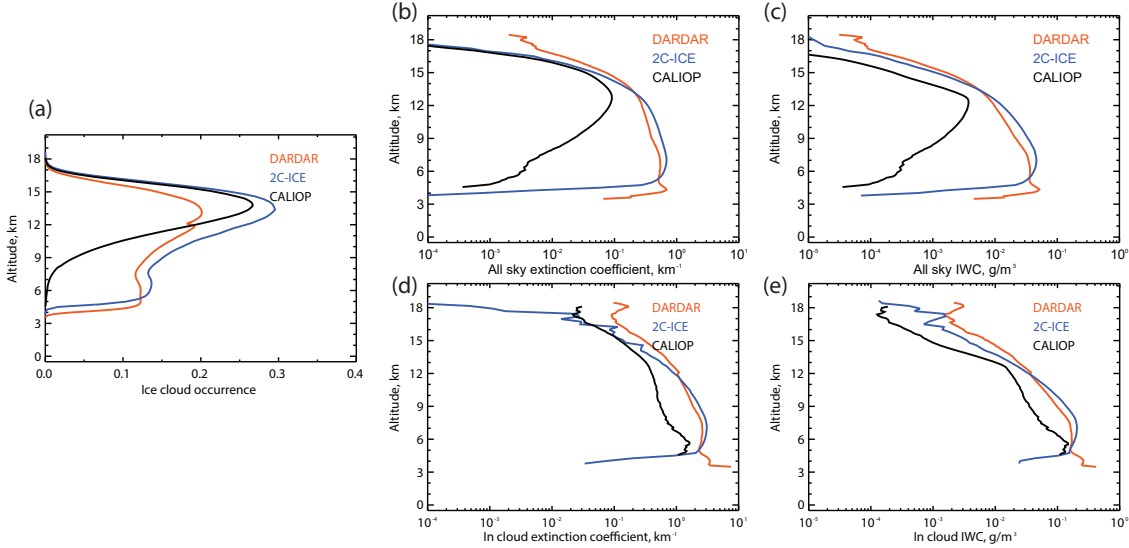

**Figure 19.** Comparison of ice cloud occurrence (a), extinction coefficient (b, d) and IWC (c, e) profiles from L3-ICE, DARDAR, and 2C-ICE radar/lidar products in the northern tropics ($0°$ - $30°$ N) for July 2008. Mean all-sky extinction coefficient and IWC plotted in (b) and (c) are calculated using Equation 6. Mean in-cloud extinction coefficient and IWC plotted in (d) and (e) are calculated using Equation 5.



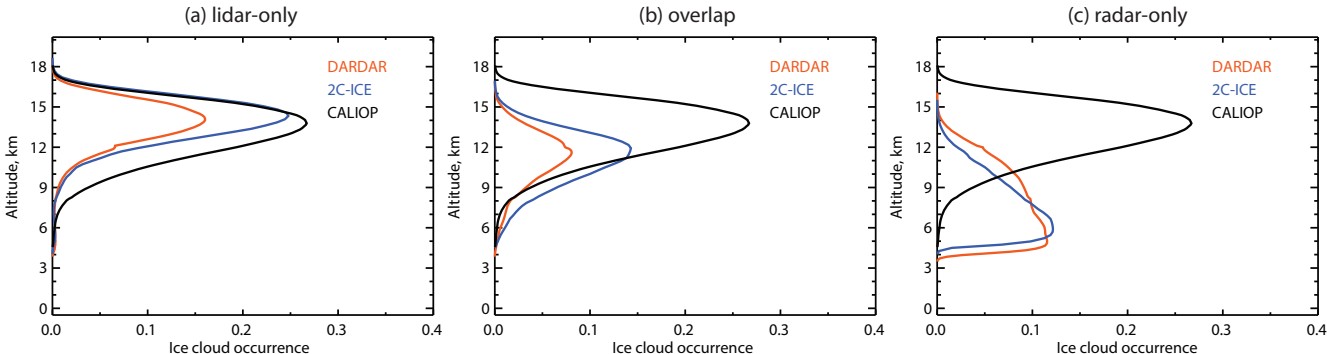

**Figure 20.** Profiles of cloud occurrence from DARDAR and 2C-ICE shown in Figure 19 (a) partitioned into lidar-only, overlap, and radar-only regions and compared with the cloud occurrences reported in L3-ICE. The profile of ice cloud occurrence from L3-ICE is included for reference in each plot.

but it is notable that there are significant differences between all three products in ice cloud occurrence frequencies at most altitudes. Below 13 km, all-sky extinction coefficients and IWC from DARDAR and 2C-ICE agree reasonably well with each other but values from L3-ICE are significantly less. These smaller all-sky values of L3-ICE extinction coefficient and IWC are driven by blockage of the lidar signal in optically thick clouds as can be seen in Figure 19 (c), where the lidar is able to detect many fewer clouds than the radar at lower altitudes. Above 12 km, however, cloud occurrence from L3-ICE is higher than from

DARDAR but similar or somewhat lower cloud than from 2C-ICE. The all-sky IWC profiles show similar features except that DARDAR and 2C-ICE both have larger IWC at high altitudes than L3-ICE and agree relatively well at all altitudes, in spite of DARDAR and 2C-ICE cloud occurrence being noticeably different at intermediate altitudes (Figure 19 (a)).

    Figures 19 (d) and (e) compare in-cloud profiles of extinction and IWC from the three products. This removes the differences between the all-sky extinction coefficient and IWC profiles which are due to differences in cloud occurrence. While significant

differences are still present, values from L3-ICE below 10 km are only a factor of 3 to 5 lower than the other two products. This remaining difference may be due, at least in part, to the ice clouds at these altitudes which are blocked from being viewed by CALIOP, but retrieved by radar, are denser than the ice clouds that are seen by CALIOP.

    For insight into the relative capabilities of CALIOP and CloudSat, Figure 20 partitions the ice cloud occurrence profiles of Figure 19 (a) into profiles of ice cloud occurrence measured within the lidar-only, overlap, and radar-only regions of the

DARDAR and 2C-ICE retrievals. The CALIOP occurrence profile of Figure 19 (a) is duplicated in each panel of Figure 20, for reference. While clouds detected by CALIOP extend to 18 km, virtually no clouds are detected by CloudSat above 14 km (Figure 20 (c)). Lidar-only 2C-ICE cloud occurrences above 15 km agree well with L3-ICE, while lidar-only occurrences in DARDAR are significantly lower (Figure 20 (a)). Figure 20 (c) shows the CALIOP backscatter signal can be completely attenuated as high as 15 km, in the dense tops of developing deep convective clouds while, in the northern tropics, very few

ice clouds are reported by CALIOP below 6 km.

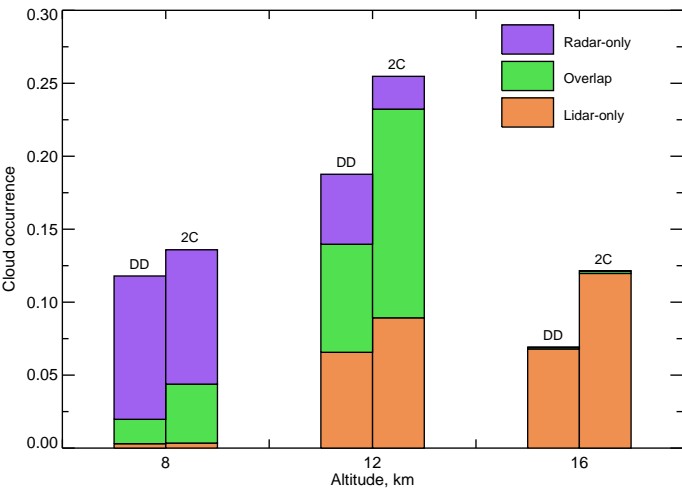

**Figure 21.** Fraction of ice samples at 8 km, 12 km, and 16 km, $0° - 30°$ N in July 2008, which fall into the lidar-only, overlap, and radar-only regions of DARDAR (DD) and 2C-ICE (2C).

**Table 6.** Tabulated fractional ice cloud occurrences corresponding to results shown in Figure 21.

|  | 8 km | | | | 12 km | | | | 16 km | | | |
|---|---|---|---|---|---|---|---|---|---|---|---|---|
|  | All | Lidar only | Overlap | Radar only | All | Lidar only | Overlap | Radar only | All | Lidar only | Overlap | Radar only |
| DARDAR | 0.119 | 0.00300 | 0.0167 | 0.0982 | 0.189 | 0.0656 | 0.0741 | 0.0480 | 0.0691 | 0.0679 | 0.00105 | 0.0000723 |
| 2C-ICE | 0.136 | 0.00345 | 0.0404 | 0.0921 | 0.255 | 0.0893 | 0.143 | 0.0225 | 0.121 | 0.120 | 0.00164 | 0.0000355 |

Figure 21 quantifies results shown in Figure 20 by showing the relative number of ice cloud samples from DARDAR and from 2C-ICE which fall in the lidar-only, radar-only, or overlap regions at three specific altitudes: 8 km, 12 km, and 16 km. The corresponding tabulated results are shown in Table 6. From Figure 21, most of the clouds at 16 km are observed by lidar only but there are some observations falling in the overlap region and a very few observations falling in the radar-only region, likely from deep convection penetrating above 16 km. About half the data at 12 km falls in the overlap region with a significant number of samples which fall in the lidar-only and radar-only regions. At 8 km, most of the data samples are in the radar-only region with very few lidar-only samples.

Figure 22 gives additional insights into the differences between products, showing histograms of IWC at the same three altitudes near the equator (10° S to 10° N). The histograms are computed from DARDAR and 2C-ICE in the same manner as for L3-ICE. The legend shows the mean in-cloud IWC from each product, computed using Equation 5 with mid-bin values. All the histograms of IWC shift toward larger values of IWC as altitude decreases. At 16 km (Figure 22 (a)), dominated by





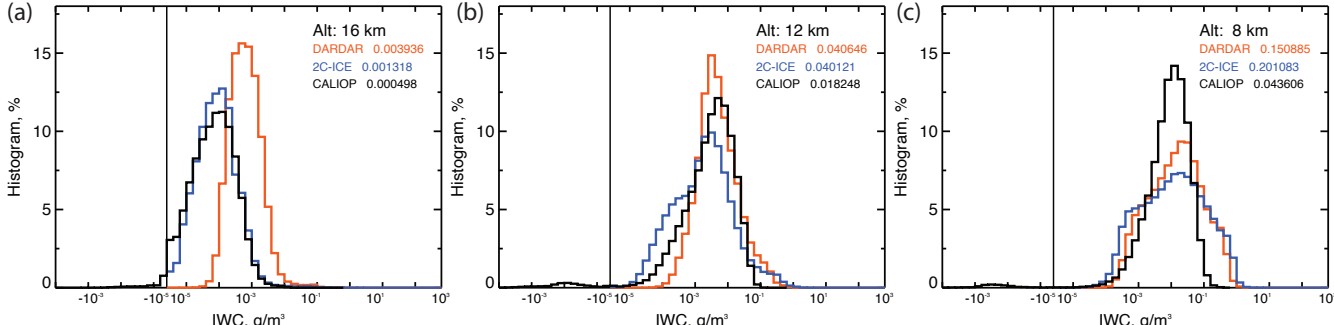

**Figure 22.** Normalized histograms of IWC at altitudes of a) 16 km, b) 12 km, and c) 8 km near the equator ($10°$ S - $10°$ N) in July 2008 from the L3-ICE, DARDAR, and 2C-ICE products, plotted in black, orange, and blue, respectively. Negative IWC values from L3-ICE are included on the left of the black solid vertical line.

lidar sampling (Figure 21), the L3-ICE and 2C-ICE histograms agree quite well, while the DARDAR histogram is shifted to larger values. At 12 km (Figure 22 (b)) and at 8 km (Figure 22 (c)), L3-ICE reports very little IWC greater than $0.1 \, \mathrm{g\,m^{-3}}$ while DARDAR and 2C-ICE report a significant number of larger values contribute to an IWC which is 3 to 5 times higher than from L3-ICE.

## 7 Discussion and summary

That significant difficulties in the simulation of tropospheric ice clouds in climate models are due in part to a lack of high quality, global observations was noted more than a decade ago (Waliser et al., 2009), with perhaps the most critical need being for vertically resolved observations of ice mass, and this need persists (Duncan and Eriksson, 2018). We have described a contribution to address this need, the monthly gridded CALIPSO Level 3 Ice Cloud Product (L3-ICE) derived from over ten years of global, near-continuous measurements from the CALIPSO lidar. The primary contents of the product are vertically resolved statistics on ice cloud extinction coefficients – relevant to the radiative effects of ice clouds – and profiles of cloud IWC. We have described the structure of the L3-ICE product and the methodology for its construction. Given the unique structure of the product, we have provided several examples of how to use it. We have discussed several sources of uncertainty in the product and biases due to the limited ability of lidar to penetrate optically dense clouds. Finally, we have performed a brief comparison of L3-ICE with two frequently used radar-lidar ice cloud products, finding interesting differences at high altitudes in the lidar-only region which deserve further study.

A number of studies have pointed out large discrepancies in IWP between various satellite data sets (e.g., Waliser et al., 2009; Duncan and Eriksson, 2018). Much of the difference in mean values is due to differences in the ability of sensors to retrieve large IWP values, which can often drive monthly averages. In L3-ICE we have reported the monthly statistics as histograms rather than means and standard deviations, which provides two primary advantages. First, it allows data users to choose similar parameter ranges when comparing with ice cloud products from instruments having different detection and





retrieval sensitivities. Second, the histograms are provided as sample counts to facilitate aggregation to larger space-time scales for climatological analyses or to reduce sampling uncertainties.

Our limited comparisons of L3-ICE to DARDAR and 2C-ICE illustrate the nature of differences between the products. The most obvious differences are due to the lack of observations from optically dense clouds due to reliance of 2C-ICE on lidar only. More subtle differences are seen in the lidar-only and overlap regions of DARDAR and 2C-ICE. The effective ice particle diameter, $D_e$, is a key parameter relating IWC to the active profile measurements. Dolinar et al. (2022) compared effective particle diameters ($D_e$) retrieved by 2C-ICE with the $D_e$ which is used implicitly in the L3-ICE parameterization (H14),

finding large differences. The parameterization of $D_e$ in H14 was based on the best available datasets, at the time, of airborne in situ measurements. In situ ice cloud measurements provide limited sampling of ice cloud properties, as measurements are only made along aircraft trajectories, so one might think the full-cloud profile retrievals of 2C-ICE and DARDAR in the radar-lidar overlap regions would be more representative. However, the 2C-ICE and DARDAR retrievals are critically dependent on assumptions on ice particle mass-size relationships (Cazenave et al., 2019). The mass-size relationships are only

partially constrained by the lidar-radar profile information, so they represent a potentially significant source of error. Therefore, determining whether the parameterization of H14 or the microphysical model underlying the radar-lidar retrievals provides a better retrieval of IWC is an open question at this time.

  L3-ICE contains a number of different sample counts (Table 2). Some of these are necessary to compute extinction coefficients and IWC (Section 4) while others are provided for insight into the presence of clouds of liquid or unknown phase. The

sample counts in L3-ICE are appropriate for estimating the vertical occurrence frequencies of ROI clouds but do not contain sufficient information to reliably characterize cloud occurrence for liquid and HOI clouds. Quality screening in L3-ICE is designed to ensure selection of high quality extinction and IWC retrievals in ice clouds and is only applied to cloud layers identified as ice. $N_{acc}$ represents the number of quality screened samples within high confidence ROI clouds and does not include ice layers identified as HOI, while $N_{ice}$ includes both ROI and HOI but has not been quality-screened. Therefore, neither

of these sample counts is an accurate representation of ice cloud occurrence. Further, sample counts for liquid clouds, $N_{liq}$, may be biased high due to inclusion of low confidence layers which represent detection artifacts rather than true cloud layers (see Figure 2 (b)). For all these reasons, cloud occurrences computed from the sample counts in L3-ICE may be different from other CALIOP level 3 cloud products. But these differences are primarily due to the different objectives and strategies used in creating the cloud products and should not be considered as characterizing the uncertainties in the products.

The current L3-ICE product (V1.00) covers the time period from June 2006 through December 2016. Increasing occurrences of CALIOP low energy laser shots have occurred since 2017. Obvious impacts on monthly Level 3 cloud statistics are seen within the South Atlantic Anomaly (SAA) by 2020, but there may be more subtle impacts as early as 2017. Therefore, CALIOP data acquired from January 2017 onwards was not included in this initial release. The CALIPSO team is currently developing an approach to mitigate the impact of these low energy laser pulses on the CALIOP Level 2 data products. A future version

of the L3-ICE product will be produced for the entire CALIOP mission period after this low energy mitigation algorithm has been applied to the Level 2 data.



## 8   Data availability

The Level 3 Ice Cloud product (Winker et al., 2018) and other CALIPSO data products are freely available from the NASA Langley Research Center Atmospheric Sciences Data Center, https://asdc.larc.nasa.gov/project/CALIPSO, and also from the ICARE/AERIS Data and Service Center, https://www.icare.univ-lille.fr/calipso/, in Lille, France. The AERIS/ICARE Data and Services Center also hosts the DARDAR data product (Delanoë et al., 2023). The 2C-ICE product (Deng et al., 2019) is available from the CIRES data center, https://www.cloudsat.cira.colostate.edu/data-products/2c-ice.

**Appendix A:  Science Data Set (SDS) of the CALIOP L3 ice cloud product**

| Name | Definition |
|---|---|
| Longitude_Midpoint | Longitude at the grid cell midpoint |
| Latitude_Midpoint | Latitude at the grid cell midpoint |
| Altitude_Midpoint | Altitude at the grid cell midpoint |
| Extinction_Coefficient_532_Bin_Boundaries | Bin boundaries used for ice cloud total extinction coefficient (channel 532 nm) histogram |
| Ice_Water_Content_Bin_Boundaries | Bin boundaries used for ice water content histogram |
| Pressure_Mean | Mean of all pressures reported within the latitude/longitude/altitude grid cell derived from the Modern Era Retrospective-Analysis for Research (MERRA-2) reanalysis product |
| Pressure_Standard_Deviation | Standard deviation of all pressures reported within the latitude/longitude/altitude grid cell from the MERRA-2 |
| Temperature_Mean | Mean of all temperature reported within the latitude/longitude/altitude grid cell from the MERRA-2 |
| Temperature_Standard_Deviation | Standard deviation of all temperature reported within the latitude/longitude/altitude grid cell from the MERRA-2 |
| Relative_Humidity_Mean | Mean of all relative humidity reported within the latitude/longitude/altitude grid cell from the MERRA-2 |
| Relative_Humidity_ Standard_Deviation | Standard deviation of all relative humidity reported within the latitude/longitude/altitude grid cell from the MERRA-2 |
| Tropopause_Height_Mean | Mean of all tropopause height reported within the latitude/longitude grid cell from the MERRA-2 |
| Tropopause_Height_Standard_Deviation | Standard deviation of all tropopause height reported within the latitude/longitude grid cell from the MERRA-2 |



| Name | Definition |
| --- | --- |
| DEM_Surface_Elevation_Minimum | Minimum of surface elevation for all columns reported in the latitude/longitude grid cell above local mean sea level obtained from the GTOPO30 digital elevation map (DEM). |
| DEM_Surface_Elevation_Maximum | Maximum of surface elevation for all columns reported in the latitude/longitude grid cell from the DEM |
| DEM_Surface_Elevation_Median | Median of surface elevation for all columns reported in the latitude/longitude grid cell from the DEM |
| Land_Surface_Samples | Number of 5 km columns within the latitude/longitude grid cell having surface type at the middle-point lidar footprint classified as land (all surface types other than water) by the International Geosphere-Biosphere Programme (IGBP) |
| Water_Surface_Samples | Number of 5 km columns within the latitude/longitude grid cell having surface type at the middle-point lidar footprint classified as water by the IGBP |
| Days_Of_Month_Observed | The days of month observed flags are bit-mapped 32-bit floats indicating which calendar days of the month CALIPSO made observations within a latitude/longitude grid cell |
| Extinction_Coefficient_532_Histogram | Histogram of ice cloud extinction coefficient derived from the 532 nm channel in the latitude/longitude/altitude grid cell |
| Ice_Water_Content_Histogram | Histogram of ice cloud content in the latitude/longitude/altitude grid cell |
| Extinction_Coefficient_532_Median | Median ice cloud extinction coefficient derived from the 532 nm channel in the latitude/longitude/altitude grid cell |
| Ice_Water_Content_Median | Median ice water content in the latitude/longitude/altitude grid cell |
| Lidar_Surface_Subsurface_Samples | Number of lidar-detected surface or subsurface samples in the latitude/longitude/altitude grid cell |
| Totally_Attenuated_Samples | Number of totally attenuated samples in the latitude/longitude/altitude grid cell |
| Cloud_Free_Samples | Number of cloud free samples in the latitude/longitude/altitude grid cell |
| Cloud_Samples | Number of cloud samples in the latitude/longitude/altitude grid cell |
| Water_Cloud_Samples | Number of water cloud samples in the latitude/longitude/altitude grid cell |
| Unknown_Cloud_Samples | Number of unknown phase or not determined cloud samples in the latitude/longitude/altitude grid cell |
| Ice_Cloud_Samples | Number of ice cloud samples in the latitude/longitude/altitude grid cell |





| Name | Definition |
|------|------------|
| Ice_Cloud_Rejected_Samples | Number of ice clouds samples failed to pass the quality control filters in the latitude/longitude/latitude grid cell |
| Ice_Cloud_Accepted_Samples | Number of ice clouds samples passed the quality control filters in the latitude/longitude/latitude grid cell |

## Appendix B:  Metadata of the CALIOP L3 ice cloud product

| Name | Definition |
|------|------------|
| Product_ID | Data product name |
| Date_Time_of_Production | Date at granule production |
| Nominal_Year_Month | Year and month when data within the Level 3 file was measured by CALIPSO |
| Program_Configuration | Contents of the runtime program configuration file which controls the sizes of dimension used by the program, the cloud quality filter parameters, and an input file check |
| Number_of_Level2_Files_Analyzed | Number of Level 2 ganules analyzed to generate this Level 3 ice cloud file |
| Number_of_Bad_Profiles | Number of 5 km bad profiles excluded from aggregation |
| List_of_Input_Files | List of included granules of Level 2 5-km Cloud Profile data product to generate this Level 3 ice cloud file |

*Author contributions.*  DMW conceived and led the effort. XC, BM, and BG developed the product with contribution from AG, MA, and MV. XC performed the analysis. DMW, XC, and MV wrote the manuscript. All authors commented on draft versions of the manuscript.

*Competing interests.*  The authors declare that they have no conflict of interest.

*Acknowledgements.*  This work was supported by NASA SMD through the CALIPSO project. We thank James Campbell, Erica Dolinar, and Andy Heymsfield for helpful discussions on the intercomparison of CALIOP retrievals with those from DARDAR and 2C-ICE. We thank
the AERIS/ICARE Data and Services Center for providing access to the DARDAR data and the Cooperative Institute for Research in the Atmosphere (CIRA) for providing access to the 2C-ICE data used in this study. DMW thanks Robert Pincus for fruitful discussions at the Max Planck Institute in Hamburg which helped shape the design of this product.



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
