# Peer review of "A Level 3 Monthly Gridded Ice Cloud Dataset Derived from 12 Years of CALIOP Measurements"

_Earth System Science Data, 2023_

## Referee Comment (RC1)

Review of "A Level 3 Monthly Gridded Ice Cloud Dataset Derived from a Decade of CALIOP Measurements", by Winker and coauthors, MS No.: essd-2023-373.

This is an outstanding article that describes the 1) processing of CALIOP Level 3 monthly gridded ice cloud data, 2) the use of 10 years of data to characterize the global distribution of clouds through an optical depth of about 2, 3) its limitations, and 4) the comparison of CALIOP processed data to two other retrieval algorithms that use a combination of CALIOP and CloudSat radar data. The figures are very informative and very well done. I highly recommend acceptance of the article with minor suggestions and corrections.

Eq. (1). In the article cited, was $\rho$ included in their derivation? I thought that an effective density, derived from direct measurements of the ice water content, was used in the development of the temperature-dependent equations.

What is the effect of contrails on the summary of CALIOP data, especially in the northern hemisphere?

Line 234, Figures 4 and 20. Shouldn't negative extinction coefficient values be rejected, as would be negative IWC values? Likewise, for the bins with negative IWC values.

Section 4.2, specifically ice cloud occurrence. Perhaps a better descriptor would be ice cloud fraction.

Figure 12. I'm not quite clear about the results shown, because the IWC generally decreases with temperature. Perhaps a bit more discussion would be helpful.

Section 6. This is a very interesting way to evaluate CALIPSO data utility by comparing the products to two products which retrieve most of the vertical column using a combination of lidar and radar.

Lines 440-442 You may want to mention that attenuation of the CloudSat W band radar can be significant, leading to errors in the DARDAR and 2C-ICE products.

Line 112. IWC directly measured

142: "underlying clouds"

Figure 10 is really interesting as it shows the changes due to the view angle of CALIOP.

Fig. 17. I don't see the black dots.

Figure 21 is very informative and insightful.

---

## Author Comment (AC1)

**Anonymous Referee #1**, 13 Dec 2023

*This is an excellent overview of the CALIPSO CALIOP lidar and very useful information on the global distribution of high clouds. This is an outstanding article that describes the 1) processing of CALIOP Level 3 monthly gridded ice cloud data, 2) the use of 10 years of data to characterize the global distribution of clouds through an optical depth of about 2, 3) its limitations, and 4) the comparison of CALIOP processed data to two other retrieval algorithms that use a combination of CALIOP and CloudSat radar data. The figures are very informative and very well done. I highly recommend acceptance of the article with minor suggestions and corrections.*
***Citation****: https://doi.org/10.5194/essd-2023-373-RC1*

The authors thank the reviewer for the outstanding appraisal and encouragement. The paper is intended to provide an insightful overview of CALIOP Level 3 Ice Cloud Product to help users better understand and use this product.

1. *Eq. (1). In the article cited, was $\rho$ included in their derivation? I thought that an effective density, derived from direct measurements of the ice water content, was used in the development of the temperature-dependent equations.*

   Yes. In Heymsfield et al. (2014), the density of solid ice $\rho$ , which is set to 0.91 g/cm3, is included in the derivation.

2. *What is the effect of contrails on the summary of CALIOP data, especially in the northern hemisphere?*

   This is an interesting question but is outside the scope of this paper. We have not tried to discriminate contrails from natural cirrus in the CALIPSO Ice Cloud Product but several studies have used CALIOP data to investigate contrails. For example:

   - Marjani et al., 2022: "Satellite Observations of the Impact of Individual Aircraft on Ice Crystal Number in Thin Cirrus Clouds", *Geophys. Res. Lett.*, 49, https://doi.org/10.1029/2021GL096173.
   - Iwabuchi et al., 2012: "Physical and optical properties of persistent contrails: Climatology and interpretation", *J. Geophys. Res.*, 117, https://doi.org/10.1029/2011JD017020

3. *Line 234, Figures 4 and 20. Shouldn't negative extinction coefficient values be rejected, as would be negative IWC values? Likewise, for the bins with negative IWC values.*

While negative extinction and IWC values are nonphysical, they appear in the CALIPSO data product as a result of signal noise. As explained in Section 3.2, negative extinction coefficients can be found in weakly backscattering layers when background signal noise is large. Rejecting those negative extinction coefficients or IWC values would result in a positive bias when calculating mean and medians. Therefore, they should be retained.

4. *Section 4.2, specifically ice cloud occurrence. Perhaps a better descriptor would be ice cloud fraction.*

   To avoid confusion, ice cloud occurrence/fraction is replaced by "ice cloud occurrence frequency".

5. *Figure 12. I'm not quite clear about the results shown, because the IWC generally decreases with temperature. Perhaps a bit more discussion would be helpful.*

   As shown in Eq 8, the maximum IWP reported in L3-ICE is driven by the requirement that overlying optical depth < 2 and the parameterized ratio of IWC to extinction, which decreases with temperature.  To clarify the text, we have changed "corresponding maximum IWP" to "corresponding maximum IWP based on Eq. 8" and have changed "the maximum retrievable IWP is about …" to "the maximum IWP corresponding to an overlying optical depth of 2 is about …"

6. *Section 6. This is a very interesting way to evaluate CALIPSO data quality by comparing the products to two products which retrieve most of the vertical column using a combination of lidar and radar.*

   The authors thank the reviewer for the encouraging comment.

7. *Question Lines 440-442 You may want to mention that attenuation of the CloudSat W band radar can be significant, leading to errors in the DARDAR and 2C-ICE products.*

   The authors thank the reviewer for this great comment. A sentence has been added to Line number 430: "In strong convection, however, attenuation of CloudSat W-band can be significant, leading to errors in the DARDAR and 2C-ICE products."

8. *Line 112. IWC directly measured*

   As suggested, "IWC measured in situ" is changed to "IWC directly measured in situ".

9. *142: "underlying clouds"*

No change. Optically thick clouds overlying (above) the black areas are responsible for completely attenuating the lidar return signal.

*10. Figure 10 is really interesting as it shows the changes due to the view angle of CALIOP.*

Indeed changing view angle of CALIOP led to more detection of randomly orientated ice cloud layers.

*11. Fig. 17. I don't see the black dots.*

In the revision, a new color scheme is utilized to make black dots standout.

*12. Figure 21 is very informative and insightful.*

The authors thank the reviewer for the good comment.

---

## Author Comment (AC2)

**Anonymous Referee #2**, 09 Jan 2024

*The authors make a laudable contribution to global ice cloud measurements through a useful and well-documented data product with valuable comparisons to peer data products. Especially useful for users are formulae and examples of how to process the product's vertically-resolved histograms into more commonly used quantities. Below, I highlight several points where further clarification could be useful.*

The authors thank the reviewer for the great appraisal and encouragement. The paper is intended to provide an overview of the CALIPSO Level 3 ice cloud product, including the algorithm, core content, demonstration of usage and comparison with other similar products. We also appreciate the constructive feedbacks from the reviewer. The comments are addressed in detail below.

Minor comments:

1. *Lines 221-223: The authors write "Comparison of panels (e) and (f) shows the most significant impact of applying quality filters is the exclusion of bins deep within opaque cloud layers where the overlying optical depth exceeds 2, such as near latitudes 19.0◦ N and 11.0◦ N." This sentence is confusing since it refers to overall product behavior anecdotally via the example of a single granule -- the "exclusion of bins deep within opaque layers" is a general statement about the product while "near latitudes 19°N and 11°N" is a particular statement about the example.*

   To avoid confusion, this sentence is revised as *"Comparison of panels (e) and (f) shows the most significant impact of applying quality filters is the exclusion of bins deep within opaque cloud layers where the overlying optical depth exceeds 2, as demonstrated near latitudes 19.0º N and 11.0º N in this case study."*

2. *Lines 263-265: The authors write "In L2-CPro, the AVD is reported at 60 m vertical resolution between 8.2 km and 20.2 km but reported at 30 m vertical resolution below 8.2 km, while the vertical resolution of L3-Ice is 60 m at all altitudes." However, the reported resolution of L3-Ice is 120 m as documented elsewhere in the text. Perhaps the authors could clarify how this 60 m resolution appears to be an intermediate aggregating resolution rather than a final output resolution.*

   In the first draft, the authors provided a brief explanation at Line 267: *"When aggregating two 60-m bins to one L3 120-m vertical bin, each 60-m cloudy bin is considered as one sample count."* In the revised version, the authors further provided an analogy with aggregation method of passive sensors to help clarify: *"When aggregating two 60-m bins to one L3 120-m vertical bin, each 60-m cloudy bin is considered as one sample count thus the sample count in this L3 120-m bin would be two. This aggregation method from fine vertical bin to coarse vertical bin*

*is analogous to accumulating level 2 passive sensor data with a spatial scale of tens of kilometer, for example, into a level 3 coarse horizontal grid such as 1.0º longitude by 1.0º latitude."*

3. *Figures 7 & 8: I am curious about the small but noticeable elevated IWC and extinction at 4 km visible at most latitudes, appearing as a horizontal band, which is also noticeable in the percent of ice samples rejected. It looks as if it might be an effect of either the CPro retrieval or QC filtering, but I did not see it discussed in the text.*

The elevated IWC and extinction coefficient around 4 km is an artificial discontinuity introduced by the single shot clearing algorithm, which deserves a detailed analysis in a future study. However as ice clouds are mainly formed above 4 km, the impact of this artificial discontinuityon the IWC and ice cloud extinction coefficients is probably small.

Still users should be aware of this potential impact. In the revised version, a new paragraph is added at the end of Section 4.2. It is also included here.

*"It is noticed that a discontinuity appears around 4 km in mean zonal IWC and extinction coefficient patterns in Figure 7 and the percentage of removed ice cloud samples in Figure 8. This discontinuity is likely due to the boundary layer cloud-clearing process in the Level 2 feature detection algorithm. More details on the boundary layer cloud-clearing process are provided in Section 5.3. "*

4. *Line 403-407: I found the impact of cloud-clearing on scattering/extinction fields somewhat vague. Are the authors stating that, for intermittently cloudy layers at 5 km resolution with <4 km top height, the resulting cloudy extinction/IWC used by L3-ICE ignores scattering from single-shot-detectable clouds?*

Yes, that is correct.  Due to an oversight in the design of the Level 2 algorithms, the cloud extinction coefficients and IWC of clouds detected in single shots is ignored in constructing the Level 2 CPro product.  This then propagates into L3-ICE. The impact of this clearing of clouds detected in single shots is difficult to quantify, however, due to the varied impacts of removing strongly scattering clouds from the 5-km averages.  An investigation showed that IWC in L3-ICE could be either increased or decreased by cloud clearing, depending on the details of the circumstances.  In this section we are only trying to point out the existence of this source of uncertainty and where, geographically, it is a concern.

5. *Section 6: In the comparison between L3-ICE and the combined radar-lidar products, many differences are stated to be "significant," but I do not see any statistical significance testing in this section.*

The sample size of the data used to produce these plots is large, so even small differences would be statistically significant.  But the word "significant" is used here in the sense of an observed difference which is clearly noticeable to the eye.

6. *Radar-lidar vs. lidar-only comparison: The authors provide an invaluable comparison to similar products but provide relatively little discussion of large differences in ice cloud frequency in the region where one would expect the greatest agreement (>15 km/lidar-only). One product agrees well with L3-ICE, while another (DARDAR) shows far fewer ice clouds (Fig. 20a). Do the authors expect L3-ICE ice cloud occurrence to be more accurate for high clouds than the combined products, or do the authors think definitional differences could explain such a large discrepancy?*

    The cloud mask algorithms associated with L3-ICE, DARDAR, and 2C-ICE are quite different.  Each of these algorithms involve several adjustable parameters.  Parameter values are typically chosen to optimize the ability of the algorithm to address a particular set of objectives.  How these parameters are "tuned" to meet the objectives will have an effect on the frequency of the reported cloud occurrence. The 2C-ICE cloud mask algorithm, which operates quite differently from the CALIOP algorithm, seems to have been tuned to agree with the CALIOP cloud mask.  DARDAR was developed for application to EarthCARE joint radar-lidar observations (Ceccaldi et al. 2013), using CALIPSO-CloudSat to develop a prototype algorithm.  EarthCARE has a requirement to provide information on cloud occurrence and composition at a 1-km horizontal scale.  Because of this the DARDAR algorithm does not do the extensive averaging used by the CALIOP algorithm and so has less sensitivity to the optically thin cirrus which is prevalent in the tropical upper troposphere.  Analysis of CALIOP cloud data shows the difference between CALIOP and DARDAR near 15 km in Figure 20 (a) is, qualitatively, what one would expect from limited sensitivity to thin cirrus.

*Text/figure corrections:*

The authors thank the reviewer taking extra efforts to improve the paper. The suggested corrections have been implemented in the revised version.

7. *Figure 5: caption reads "Level 3Tropospheric" (missing a space)*

    Missing space is now added between "Level 3" and "Tropospheric".

8. *Figure 7: units of IWC and ice extinction are not specified in either the figure or the caption.*

    Units of IWC (g m$^{-3}$) and ice extinction (km$^{-1}$) have been added in the caption.

9. *Figure 11: Meaning of grey pixels in both panels is not stated.*

Grey pixels represent averaged surface and subsurface at each latitude detected either from CALIOP lidar or DEM if signals are totally attenuated. The explanation now is added in the caption of Figure 11:

" Mean surface and the subsurface below  determined from lidar detection, or from the DEM if the signal has been fully attenuated, are shaded in gray; …"

10. *Lines 287-298: "all-sky IWC (Figure 7 (h) and (j) shows a rainbow-shaped maximum" -- missing ")"*

The missing ")" is now added in the revision.

11. *Line 386: "when averaged using a 10◦×10◦ (red))" -- extra ")"*

The extra ")" has been removed in the revision.

12. *Line 408: "impacts L2 CPro ice cloud extinction" -- elsewhere it's spelled as "L2-CPro"*

The "L2 CPro" is replaced by "L2-CPro" to ensure the consistency of the paper.

13. *Line 504: "L3-ICE reports very little IWC greater than 0.1 gm−3 while DARDAR and 2C-ICE report a significant number of larger values contribute to an IWC which is 3 to 5 times higher than from L3-ICE." -- grammatically confusing*

The sentence is replaced by "L3-ICE reports very few occurrences of IWC greater than 0.1 gm−3 while DARDAR and 2C-ICE identify a significant number of IWC occurrences above this threshold, which results in a much smaller average IWC than that from the DARDAR or 2C-ICE histograms."

14. *Line 562: the CloudSat data center is at CIRA (CSU/NOAA), not CIRES (CU Boulder/NOAA)*

The CloudSat data center is now corrected.